# Accumulation of soil organic C and N in planted forests fostered by tree species mixture

Yan Liu[1], Pifeng Lei[1,2,*], Wenhua Xiang[1], Wende Yan[1,2], Xiaoyong Chen[2,3]

[1] Faculty of Life Science and Technology, Central South University of Forestry and Technology, Changsha, Hunan, 410004, China

[2] National Engineering Laboratory for Applied Technology of Forestry & Ecology in South China, Central South University of Forestry and Technology, Changsha 410004, Hunan, China

[3] Division of Science, College of Arts and Sciences, Governors State University, University Park, Illinois 60484, USA

*Correspondence to:* Pifeng Lei (pifeng.lei@outlook.com)

**Abstract:** With the increasing trend of converting monocultures into mixed forests, more and more studies were carried out to investigate the admixing effects on the tree growth and aboveground carbon storage. However, few studies have considered the impact of mixed forests on the belowground carbon sequestration, and changes of soil carbon and nitrogen stock as forest grows in particular. In this study, paired pure *Pinus massoniana* plantations*, Cinnamomum camphora* plantations and mixed *Pinus massoniana -Cinnamomum camphora* plantations at ages of 10, 24 and 45 years were selected to test whether the mixed plantations sequestrate more organic carbon (OC) and nitrogen (N) in soils and whether this admixing effect becomes more pronounced with stand ages. The results showed that tree species identity and composition as well as stand age significantly affected soil OC and N stock. The soil OC and N stocks were the highest in mixed *Pinus-Cinnamomum* stands compared to those in counterpart monocultures within the same age in the whole soil profiles or specific soil depth layers (0-10 cm, 10-20 cm and 20-30 cm) for most cases, followed by *Cinnamomum* stands, and the *Pinus* stands the lowest. And these positive admixing effects were mostly non-additive. Along chronosequence, the soil OC stock peaked at 24-yr-old stand and maintained relatively stable thereafter. The admixing effects were also the highest at this stage. However, at topsoil layer, the admixing effects increased with stand ages in terms of soil OC stocks. When comparing mixed *Pinus-Cinnamomum* plantations with corresponding monocultures within the same age, the soil N stock in mixed stands was 8.30%, 11.17% and 31.45% higher than the predicted mean value estimated from counterpart pure species plantations in 10-yr-old, 24-yr-old and 45-yr-old stands, respectively, suggesting these admixing effects were more pronounced along chronosequence.

# 1 Introduction

Soil carbon is more stable than that stored in plants, which makes soil carbon more resistant to disturbance (Cunningham et al., 2015). Organic carbon from forest soils accounts for 70%-73% global soil organic carbon (Six et al., 2002), and there is an adorable part of the global forests being plantations. Thereby, plantations play an important role on soil carbon sequestration and mitigating global atmospheric carbon budget (Cunningham et al., 2015). However, considering the problems caused by pure plantations (Peng et al., 2008), nowadays planting mixed forest with different species in one stand is becoming more and more popular in worldwide plantation management (Felton et al., 2016; Oxbrough et al., 2012), for the potential positive effects (Knoke et al., 2007). Since mixed forests, compared with monocultures, were generally characterized by the sustainability to resist disturbance, potential of higher yield and better ecological services (Grime, 1998; Knoke et al., 2007). It is generally accepted that mixed forests exert favorable effects over monocultures by two main mechanisms-complementary effect and selection effect (Isbell et al., 2009; Grossiord et al., 2014). The former is explained by inducing facilitation or ease interspecific competition and enhancing resource use efficiency by niche partitioning, and the latter by increasing the possibilities of including highly productive species to increase the yield. Though a lot of work has been done to study positive effects of biodiversity to ecological function (Balvanera et al., 2006; Marquard et al., 2009), while the attention is rarely being attached to soil organic carbon (Vesterdal et al., 2013) and nitrogen stocks, which are closely related to global climate change. The limited researches showed that the soil OC and N stocks in mixed stands do not necessarily exceed that of corresponding pure stands (Berger et al., 2002; Forrester et al., 2012; Wiesmeier et al., 2013; Wang et al., 2013; Cremer et al., 2016), while most of them did not consider the relative portion of the component species (Berger et al., 2002; Wang et al., 2013; Cremer et al., 2016), which may result in underestimation of admixing effects. For example, admixing effect could also exist, because of the small portion of the higher-production species which leads to low expected production, even if the production of two-species mixed stand is between that of two corresponding pure stands.

The studies of monocultures suggested that soil organic carbon stock of the upper soil layer fluctuated in the early stage of afforestation until it reached a new equilibrium which depends on the rates of litter input and decomposition (Paul et al., 2002; Tremblay et al., 2006; Sartori et al., 2007). And for soil nitrogen stock, it increases with increasing stand age in the upper soil layer because of the accumulative biological fixation, litterfall, recycling from deeper mineral soil via root mortality, and even atmospheric N deposition (Hume et al., 2016). However, the changes of soil organic carbon and nitrogen stocks at different stand ages are rarely quantified and poorly understood, and it is especially true in the context of comparison of pairwise mixed forests with pure forest stands along chronosequence.

In this study we investigated the soil organic carbon and nitrogen stocks of *Pinus massoniana* and *Cinnamomum camphora* pure stands as well as their mixed stands at the 10, 24 and 45 years old. All the soil samples were taken from 0-10, 10-20 and 20-30 cm soil of above stands in Hunan province in China. We hypothesized that (1) the soil organic carbon and nitrogen stocks under mixed stands are higher than those under corresponding monocultures within the same age in whole soil profile;

(2) these positive admixing effects become more pronounced along chronosequence.

## 2 Materials and methods

### 2.1 Study site

Three parallel forest stands at different stand ages were selected, comprising mixed *P. massoniana* and *C. camphora* plantations, pure *P. massoniana* plantations and pure *C. camphora* at 10, 24 and 45 years old. Therein some *Pinus elliotii* were spotted in between and we treated it as *P. massoniana* because of the similarity of growth and biological characteristics of them (we grouped them as "*Pinus*" thereafter). Three 20 m×20 m plots were established in each of the forest stand, including mixed *Pinus-Cinnamomum* stands and corresponding pure species stands (*Pinus* and *Cinnamomum*) at age of 24 and 45 years old in March 2013 in the Botanical Garden in Hunan Province, China (28°06′N,113°02′E), amounting to 18 plots. Due to the lack of young stands in the Botanical Garden, we selected another site located in Taolin forestry station (28°55′N, 113°03′E), Hunan Province, China. This site was used as nursery and abandoned since 2000, and then planted in 2003. Three plots in each of the three plantations types at 10 years old were set up with same method mentioned above, but with size of 12 m×12 m constrained by the smaller patches there. In total, our study included 27 plots consisting of mixed *Pinus-Cinnamomum* plantations and corresponding monocultures at age of 10, 24 and 45 years old. These two sites are of 200 km in distance and with similar climate and soil type. The regional climate is typical mid-range subtropical monsoonal with mean annual air temperature of 17.2℃, mean annual precipitation of 1422 mm in Botanical Garden and mean annual air temperature of 16.9℃, mean annual precipitation of 1353.6 mm in Taolin forestry station. The soil is well-drained clay-loam red soil developed from slate parent rock, classified as Alliti-Udic Ferrosols, according to Acrisol in the World Reference Base for Soil Resources (Institute of Soil Science, Chinese Academy of Science 2001). The depth of soil layer is deeper than 80 cm, but the content of soil humus was not rich and pH value ranges from 4.12 to 4.86. The forests remained unmanaged and all soil-forming factors had remained constant since forest establishment owning to either the foundation of Botanic Garden or the young age of the forests. Here we used two sites for this experiment due to the difficulties of finding one field site with all these plantations in gradient of forest ages. The principle of field site selection here was to put three plantation types at the same age in one site. Therefore, we think these two sites are suitable for our experiment since our purpose here was to evaluate the admixing effects on soil C and N stocks by comparing mixed forests and corresponding monocultures at the same age rather than comparing soil C and N in forests at different ages. More detailed information about the experimental site and soil condition referred to Wen et al. (2014).

### 2.2 Soil sampling and laboratory analyses

Four soil samples were randomly collected in each plot using a metal corer (10 cm in diameter) from three depths: 0-10, 10-20, 20-30 cm and each of them were treated as one individual sample, then air-dried and sieved through 0.25 mm sieve.

Here we collected the soil samples down to 30 cm depth as the soils in this area were susceptible to the environmental variations and sensitive to the carbon input by litter and fine roots, which matches the purposes to assess the admixing effects on soil OC and N stock over time (Wang et al., 2013; Cremer et al., 2016). Soil organic carbon was determined by the wet combustion method using oxidization of potassium bichromate (Walkley-Black method) and then titrated with 0.5 N ferrous ammonium sulfate solution by using diphenylamine indicator. Soil total nitrogen was measured using the Semimicro-Kjeldahl method digested with a mixture of $H_2SO_4$, $K_2SO_4$, $CuSO_4$ and Se (Institute of Soil Science, Chinese Academy of Science).

## 2.3 Data analysis

Soil OC stock (t ha$^{-1}$) and N stock (t ha$^{-1}$) at different soil depth in different stands were calculated with formula: stock =bulk density* depth *[OC or N concentration]. The effects of the experimental factors, species composition, stand age and soil depth, on soil OC, N concentrations and soil OC, N stocks were tested by means of three-way analysis of variance (ANOVA). Differences in soil OC or N concentration and stock among mixed *P. massoniana* and *C. camphora* stands, pure *P. massoniana* stands and pure *Cinnamomum* stands within the same age stages and soil layers were analyzed by using one-way ANOVA, followed by Tukey's test. In order to detect whether the admixing effects were additive or non-additive, an alternative analytical method was used as suggested by Ball et al. (2008). Expected values of OC and N stock in mixed stands at different stand ages in different soil depths were calculated by adjusted values based on basal area in monocultures with formula:

$$\text{Expected value}=(Ba_{p.mix}/ Ba_{p.pure})*Stock_{p.pure} + (Ba_{c.mix}/ Ba_{c.pure})*Stock_{c.pure}$$

Where the $Ba_{p.mix}$ is the basal area of *Pinus* in the mixed stand, $Ba_{p.pure}$ is the basal area of *Pinus* in the pure *Pinus* stand, the $Ba_{c.mix}$ is the basal area of *C. camphora* in the mixed stand, $Ba_{c.pure}$ is the basal area of *Cinnamomum* in the *Cinnamomum* pure stand, $Stock_{p.pure}$ is the soil OC or N stock under pure *Pinus* stand and $Stock_{c.pure}$ is the soil OC or N stock under pure *Cinnamomum* stand. All the calculations above are conducted in the same given stand age (10-yr, 24-yr or 45-yr old). Then these expected values of OC or N stock were compared with observed values for each individual sample that were measured experimentally in mixed stands with formula: (observed-expected)/expected. For soil OC and N stock in specific soil depth at given stand ages, 95% confidence intervals (CI) were calculated with above formula. If the CIs for mixtures did not cross y = 0, the admixing effect was considered non-additive (Ball et al., 2008). Otherwise, we regard the admixing effect as additive. All the statistical analysis were conducted with statistical software R project (R 3.3.0) (R Development Core Team, 2010).

# 3 Results

## 3.1 Soil organic carbon and nitrogen stocks

The three-way ANOVA indicated the forest stand types, soil depth and their interactions exerted significant influence on the soil OC and N concentrations, while the age effect were not significant for soil OC concentration (table 2). When compared with parallel forest stand within the same ages, soil OC and N concentrations were highest in *Pinus-Cinnamomum* mixed stands for almost all the cases in the whole soil profiles or in the specific soil layers, but the significant differences were only detected in 24-yr old and 45-yr-old stands (P<0.05) (Fig. 1).

Total soil OC stock was highest in *Pinus-Cinnamomum* mixed stands (44.86 Mg ha$^{-1}$) compared to corresponding monocultures at the same stand age in the whole soil profiles investigated, followed by evergreen broad-leaved *Cinnamomum* stand (36.37 Mg ha$^{-1}$), and conifer *Pinus* stand showed the lowest values (31.51 Mg ha$^{-1}$). The significant differences were detected in 24-yr old and 45-yr-old stands, but not in 10-yr old stands (P<0.05). Along chronosequence, soil OC stocks increased with increasing stand age in these three forest types, with mean value of 30.50 Mg ha$^{-1}$, 41.96 Mg ha$^{-1}$ and 43.85 Mg ha$^{-1}$ in 10, 24 and 45 years old stands, respectively (Fig. 2). When we take a closer look at stratification distribution of the soil OC stocks in these three forest stands at different stand ages, the results showed soil OC stocks decreased significantly with increasing soil depth and the similar pattern that highest OC stock in mixed stand over monocultures were observed within given soil layers at given stand ages. The over-performance of OC stock in *Pinus-Cinnamomum* mixed stands compared to the counterpart monoculture stands within the same age were mainly attributed to the top 10 cm soil depth (Fig. 3). In 0-10 cm soil layers, OC stocks under mixed stands showed no significant differences compared with individual stands in the young stands, and then those under mixed stands significantly exceeded the *Pinus* individual stands in middle-age stands, and finally exceeded all two individual stands in the oldest stands; in 10-20 cm soil layers, both 10-yr-old and 45-yr-old stands exerted no significant differences in soil OC stock among three stand types while in 24-yr-old stands mixture had significantly higher soil OC stock than the monocultures; in the 20-30 cm soil layers mixed stands showed similar soil OC stock pattern when compared with corresponding monocultures and in 24-yr-old stands mixture showed significant higher soil OC stock than two pure *Pinus* and *Cinnamomum* stand.

Soil depth and stand type, but not stand age exerted significant effects on soil N stock in mineral soil (Table 2). Within the same stand, the soil N stocks decreased with soil depth, and among them, the superficial soil layers (0-10 cm) exhibited significantly higher soil N stocks than the other two deeper soil layers (Fig. 3). Only in 0-10 cm and 10-20cm soil layers of 24-yr old stands and 20-30 cm soil layer of 45-yr old did N stock significantly differ among stand types, and pure *Cinnamomum* and *Pinus-Cinnamomum* mixed stands always had similar N stocks, which were higher than *Pinus* stands. The N stock in topsoil of mixed stand increased along chronosequence for all stands, except in *Cinnamomum* stand, which marginally decreased from 24-yr old to 45-yr old stand. And in 10-20 cm soil layer of *Pinus* stand, N stock increased from 10-yr old to 24-yr old stand then stayed stable. While all the other soil layers didn't exert significant difference along

chronosequence within stand type (Fig.2 and 3).

## 3.2 Admixing effects on soil OC and N stocks

The relative performance of soil OC and N stocks in the *Pinus-Cinnamomum* mixed stands were calculated to determine the additive or non-additive effects for each stand at different soil layers and different stand ages by comparing observed soil OC or N stock values with expected values based on counterpart monocultures. In our study the mixing planting always exerted positive effects, as all the relative values of soil sequestrated OC and N stock were positive in all the soil profiles in mixed stands. But almost all the CIs did not cross y=0, suggesting these positive admixing effects were strongly non-additive, except the relative OC sequestration in 20-30 cm soil layers of 10-yr old stand. In the 10-yr old stand, the over-performance of soil OC sequestration was mainly attributed to the top and subsoil layers, and these positive effects were stable in the two upper soil layers, and then decreased in the deepest layer. In the 24-yr old stand, it increased with soil depth. In the 45-yr old stand, the relative percentage was the highest in the deepest soil layer (20-30 cm). Overall, the percentage of over-performance in mixture was significant higher in 24-yr-old stand than that in 10-yr-old stand and then it marginally decreased to 45-yr-old stand. For soil N stock, however, the results showed consistent pattern that the positive effects increased with increasing stand ages (Fig. 4).

## 4 Discussion

### 4.1 Soil organic carbon and nitrogen stocks

The soil organic carbon stocks under three stand types always followed the order: mixed *Pinus- Cinamonum* stand> *Cinamomum* stand ≥ *Pinus* stand, except for those in the 10-20 and 20-30cm soil layer of 10-yr old stand with slight differences. Many researchers demonstrated the species diversity or species identity effects on soil OC and N accumulation in forests and revealed that species diversity and/or abundance of dominant tree species exerted effects on OC and N stocks through carbon input by litter and fine roots (Berger et al., 2002; Guckland et al., 2009; Wang et al., 2013; Dawud et al., 2016). In the previous study of this site, the annual litterfall was also the highest in the mixed *Pinus- Cinamonum* stand, followed the *Pinus*, and *Cinnamomum* was the lowest (Xu et al., 2013). Therefore, the higher soil organic carbon under mixed stands may be attributed to the higher annual litterfall in mixed stands. While the higher soil OC in *Cinamonum* stand compared to *Pinus* is likely connected with the litter decomposition rate, as the needle leaves of *Pinus* accumulated on the forest floor with lower composition rate, and the higher litter input may not necessarily increase soil carbon content (Fontaine et al., 2004), although many previous studies also showed larger carbon stocks under coniferous species than broad-leaved species (Kasel and Bennett, 2007; Schulp et al., 2008; Wang et al., 2013). With increasing stand age, the gap of soil OC stock between *Cinamomum* and *Pinus* stand became more pronounced in whole soil profiles investigated. Furthermore, when compared with broadleaf tree species, conifer species tended to allocate much more total organic matter

production to aboveground growth, which features them as fast growing tree species and caused lower direct carbon input to soil (Cuevas et al., 1991). In this study, the highest OC stock presented in 24-yr old stands, while in *Cinamomum* stands, though not significant, carbon stock still had an increasing trend from 24 to 45 years old stands. And in mixed stands, though there being relative smaller partition of *Cinamomum*, the trend of soil OC stock is similar with *Cinamomum* stands but not in *Pinus* stand. These confirmed that broadleaf tree species are prior to conifers in the long-term growth.

For soil total N stock, mixed stand exhibited priority for almost all the cases, revealing higher soil N stocks in mixed *Pinus-Cinnamomum* stands compared to corresponding monocultures. And compared to the mean value of counterpart monoculture at the same ages, the soil N stock in mixed stands increased by 8.30%, 11.17% and 31.45%, respectively, suggesting these admixing effects were more pronounced with stand ages along chronosequence. But whether this enhanced admixing effect on soil N accumulation will continue needs further investigation. In monocultures, *Pinus* stand showed priority in earlier stage while *Cinnamomum* in the later stage in term s of soil N stocks, which is similar to the trend of soil OC stocks, but less pronounced. A large amount of soil nitrogen stored in soil organic matter, and with the decomposition of soil organic matter, nitrogen will be released and enable to be uptaken by plants or leaching, as reported in our previous study that soil nitrogen concentration positively correlated to soil organic carbon concentration (Wen et al., 2014), thereby the soil nitrogen stocks, to some degree, shared the similar patterns of soil OC stocks under the same circumstances. And as Hume et al (2016) discussed, the accumulation of soil nitrogen may be slower than carbon because nitrogen is progressively locked-up in live biomass, which likely is the explanation that the soil nitrogen stock changed less significantly with stand ages and between different stand types compared with soil OC stock here.

In the uppermost soil layer, soil organic carbon stocks under three stand types are all significantly increased from 10-yr old stands to 24-yr old stands, then became stable thereafter, but that of *Pinus* stands slightly declined from 24-yr old to 45-yr old stands without significant differences. This trend are inconsistent with previous study, which reported a decline of soil organic carbon in the topsoil (0-10cm) in certain period of time after plantation establishment (Turner and Lambert, 2000). But it's consistent with the result from a Chinese fir plantation, where the soil OC stocks increased with increasing stand ages from about 10 to 20 years old, then, stabilized (Chen et al., 2013). Most of the similar researches that focus on the changes of organic carbon stock along chronosequence suggested that soil organic carbon stock in top soil would reach a stable level after approximately 20 years (Tremblay et al., 2006; Chen et al., 2013), that can be interpreted to the formation of the balance which depends on the carbon input and organic decomposition (Hume et al., 2016). While, in the initial stage of afforestation, the changes of organic carbon storage along increasing stand age differed. The conversion of plantations into natural native forests will always present a decrease of soil organic carbon in topsoil within 10 years, and then, some will rebound and increase until reach the equilibrium level (Chen et al., 2013), the others will keep decreasing to lowest level (Turner and Lambert, 2000). In our results the soil OC and N stocks decreased significantly with increasing soil depth. However, the magnitude of over-performance in the *Pinus-Cinnamomum* mixed stands over monocultures increased with soil depth (Fig 3 and Fig. 4). Top soils under forests are always most susceptible to disturbance, and it can be directly

impacted by the carbon input through litter and fine roots which always decline along soil depth (Wang et al., 2016), and our data in the same sites also suggested fine roots mostly assembled in upper soil and fine-root overyielding occurred in the topsoil layer (0-10 cm) (see Table S1 in the Supplement). Also, in our previous study (Wen et al., 2014), the ratio of soil microbial biomass/soil organic carbon was lowest in topsoil, which may suggest lower mineralization rate in deeper soil and make the the accumulation of OC and N stocks in deeper soil along chronosequence more pronounced over topsoil layer. Here we only collected the soil down to 30 cm depth, effects of species diversity and species identity on the deeper soil merits further investigation to improve model parameter of soil OC and N processes in deep soil profile.

## 4.2 Admixing effect along chronosequence

In the two topsoil layers (0-10 cm and 10-20 cm), admixing effect on soil OC was more pronounced along chronosequence. This is likely accounted for the increasingly intensive interactions between two species along increasing stand age. And in the pure stands, the intraspecific competitions also become more intensive along chronosequence, which will block carbon sequestration because of nutrient and water limits and favor the mixed forests where the interspecific competition is relatively less intensive in general (Lei et al., 2012a). The addictive effect shown in 20-30cm soil layer under 10-yr old mixed stand, the observations were almost equal to the expected values, confirmed that the room occupied by below- and above-ground tree biomass was not big enough to exert significant interaction when the stands are relatively young or the period of soil OC accumulation processes. Our results also showed, to some degree, higher admixing effects in deeper soil layer that may attribute to the lower expected values in monocultures in deeper soil layer, which makes the (observed-expected)/expected more sensitive to the increment of observed values.

The admixing effects of soil N stock in whole profile suggested a consistent increasing trend in whole soil profile. Fine roots of mixed stands compared with pure stands' are always supposed to exploit deeper soil (Cremer et al., 2016), so they may assemble more nitrogen from deeper soil (even deeper than our samples) to upper soil. Also, Lei et al. (2012b) reported higher fine root turnover caused by higher fine root production of mixed stands which potentially increased nitrogen input in mixed stands. In addition, mixed stands are more resistant to environmental disturbances and are more capable of N retention and prevent leaching in soil (Tilman et al., 1996). This merits further studies in more diverse communities in forests to concur this pattern of higher nitrogen stock under mixed stands over monocultures and these positive effects increased along chronosequence.

## 5. Conclusions

The tree species and composition as well as stand age significantly affect soil organic carbon and nitrogen stocks. Converting pure stands into mixed stands can significantly enhance soil OC and N stock. And this positive admixing effect becomes more pronounced along chronosequence for OC only in top soil, while inconsistent trend presents in the deeper soil.

However, for soil N stock, it becomes more pronounced along chronosequence in whole soil profile. Besides, the tree species identification also affects the soil OC sequestration and N stock. In topsoil *Cimamomum* stands always contain more soil OC and N stocks than *Pinus* stands due to the different strategies of carbon and nutrient allocation and different rate of organic decomposition, but these differences are less pronounced in deeper soil layers.

## Acknowledgements

This study was sponsored by National Natural Science Foundation of China (31200346), and Introduce Talent Fund of CSUFT. We thank Forest Administration on Hunan and Forest Station of Taolin for the permission to use the site. Furthermore, we acknowledge the assistance of Fang Jiang, Yuqin Xu and Hao Yi in the field and laboratory.

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

**Table 1** Stand characteristics in pure species *Pinus* stands, pure *Cinnamomum* stands and mixed *Piuns-Cinnamomum* stands at age of 10, 24 and 45 years. (mean±standard deviation)

| Stand | Age | Species | Density (n ha$^{-1}$) | Diameter at breast height (cm) | Height (cm) | Basal area (m$^2$ ha$^{-1}$) |
|---|---|---|---|---|---|---|
| | 10 | *Pinus* | 2592 | 9.38±3.26 | 5.28±3.97 | 20.06 |
| *Pinus* stands | 24 | *Pinus* | 2050 | 14.18±4.34 | 12.86±6.52 | 35.37 |
| | 45 | *Pinus* | 600 | 21.40±5.30 | 12.47±1.88 | 22.84 |
| | 10 | *Cinnamomum* | 2708 | 7.77±2.60 | 5.99±1.25 | 14.26 |
| *Cinnamomum* stands | 24 | *Cinnamomum* | 900 | 17.02±6.52 | 13.71±2.74 | 23.46 |
| | 45 | *Cinnamomum* | 800 | 21.06±6.73 | 13.24±2.29 | 30.63 |
| | 10 | *Pinus* | 902 | 7.64±1.82 | 4.73±0.82 | 4.37 |
| Mixed | | *Cinnamomum* | 1689 | 8.14±2.81 | 7.2±0.73 | 9.83 |
| *Pinus-Cinnamomum* | 24 | *Pinus* | 267 | 19.88±5.06 | 12.35±1.64 | 7.80 |
| stands | | *Cinnamomum* | 592 | 15.27±5.92 | 11.41±3.13 | 12.45 |
| | 45 | *Pinus* | 250 | 19.69±4.10 | 12.37±2.60 | 7.91 |
| | | *Cinnamomum* | 325 | 20.94±8.54 | 13.75±2.79 | 12.91 |

**Table 2** The effects of plantation stand, stand age and soil depth on soil OC concentration, N concentration, OC and N stock. Values shown are ANOVA F values and P values. Bold font indicates significant differences at $p<0.05$.

| Factor | OC concentration | | N concentration | | OC stock | | N stock | |
|---|---|---|---|---|---|---|---|---|
| | F value | P | F value | P | F value | P | F value | P |
| stand | 24.30 | **<0.0001** | 11.89 | **<0.0001** | 36.73 | **<0.0001** | 15.17 | **<0.0001** |
| age | 0.00 | 0.9942 | 60.53 | **<0.0001** | 35.99 | **<0.0001** | 0.54 | 0.5814 |
| depth | 699.57 | **<0.0001** | 205.33 | **<0.0001** | 489.87 | **<0.0001** | 65.62 | **<0.0001** |
| stand×age | 2.07 | 0.1283 | 10.50 | **<0.0001** | 2.87 | **0.0236** | 3.92 | **0.0042** |
| stand×depth | 8.61 | **0.0002** | 2.39 | 0.0937 | 5.53 | **0.0003** | 1.15 | 0.3354 |
| age×depth | 2.00 | 0.1588 | 0.05 | 0.8163 | 16.79 | **<0.0001** | 4.18 | **0.0027** |
| stand×age×depth | 0.15 | 0.8579 | 0.73 | 0.4834 | 1.03 | 0.4142 | 0.67 | 0.7217 |

**Figure Captions**

**Fig. 1** Soil organic carbon (OC) concentration and nitrogen (N) concentration in pure *Pinus*, *Cinnamomum* and mixed *Pinus-Cinnamomum* stands in 0-10 cm, 10-20 cm and 20-30 cm soil depth at the age of 10, 24 and 45 years. Error bars indicate standard errors. Different letters indicate significant differences among different stands within the same soil profile and age stages ($p<0.05$).

**Fig. 2** Total soil organic carbon (OC) and nitrogen(N) stock in pure *Pinus*, *Cinnamomum* and mixed *Pinus-Cinnamomum* stands in 0-30 cm soil depth at the age of 10, 24 and 45 years. Error bars indicate standard errors. Different letters indicate significant differences among different stands within the same ages ($p<0.05$).

**Fig. 3** Soil organic carbon (OC) and nitrogen(N) stock in pure *Pinus*, *Cinnamomum* and mixed *Pinus-Cinnamomum* stands in 0-10 cm, 10-20 cm and 20-30 cm soil depth at the age of 10, 24 and 45 years. Error bars indicate standard errors. Different small letters indicate significant differences among different stands within the same soil profile and age stages ($p<0.05$). Different capital letters indicate significant differences among different soil layers within the same stand at given stand age ($p<0.05$).

**Fig. 4** Investigation of additive or non-additive interactions for soil OC stock (a) and N stock (b) in *Pinus- Cinnamomum* mixed stands in 0-10 cm, 10-20 cm and 20-30 cm soil depth at the age of 10, 24 and 45 years. Observed values were compared to expected values calculated as the average value in monocultures of *Pinus* and *Cinnamomum*. Error bars represent 95% CI, and mixtures for which the CIs do not cross y = 0 are considered to be significantly non-additive. Different letters indicate significant differences among different stand ages within the same soil depth profile.

Fig. 1

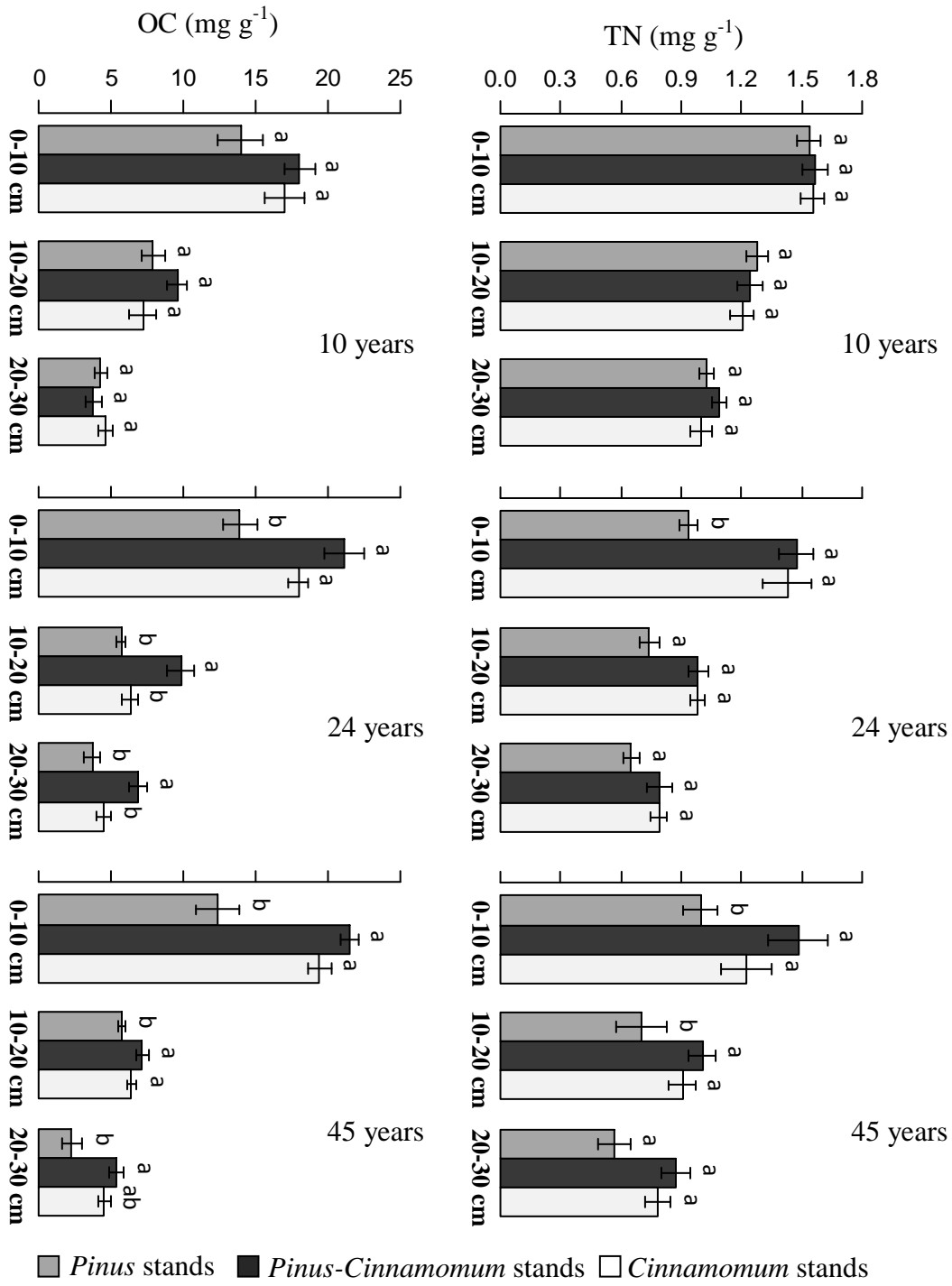

□ *Pinus* stands  ■ *Pinus-Cinnamomum* stands  □ *Cinnamomum* stands

Fig. 2

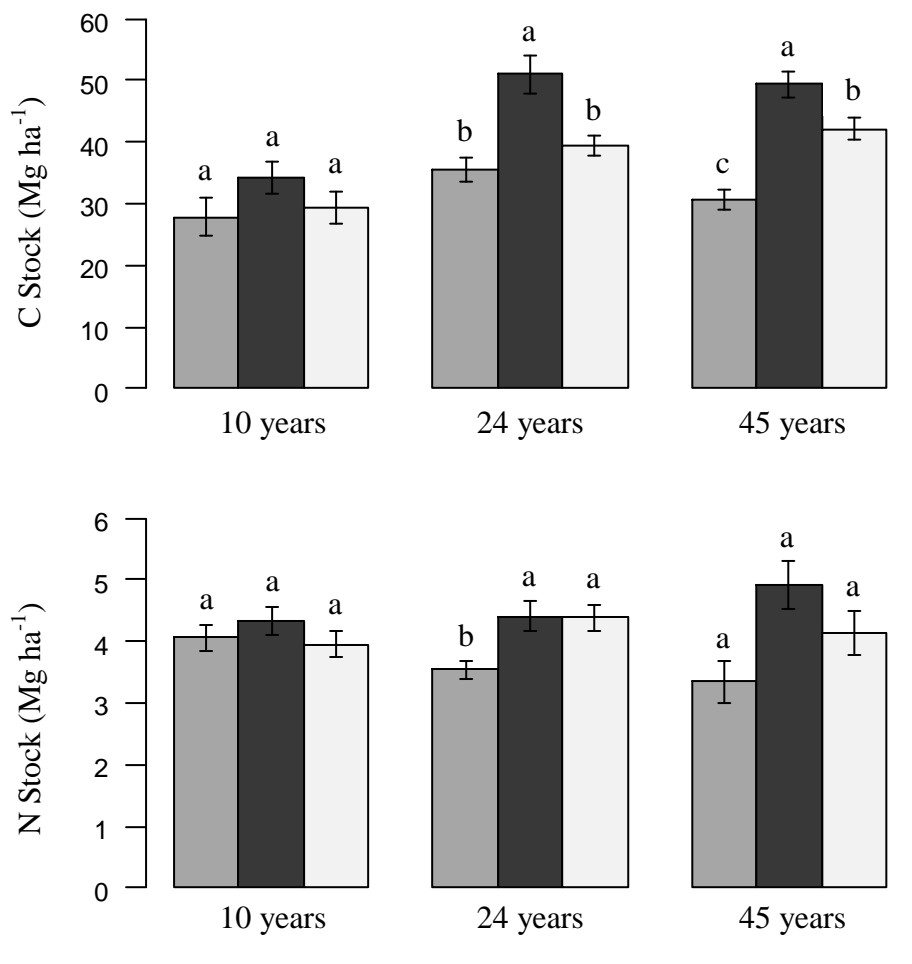

Fig. 3

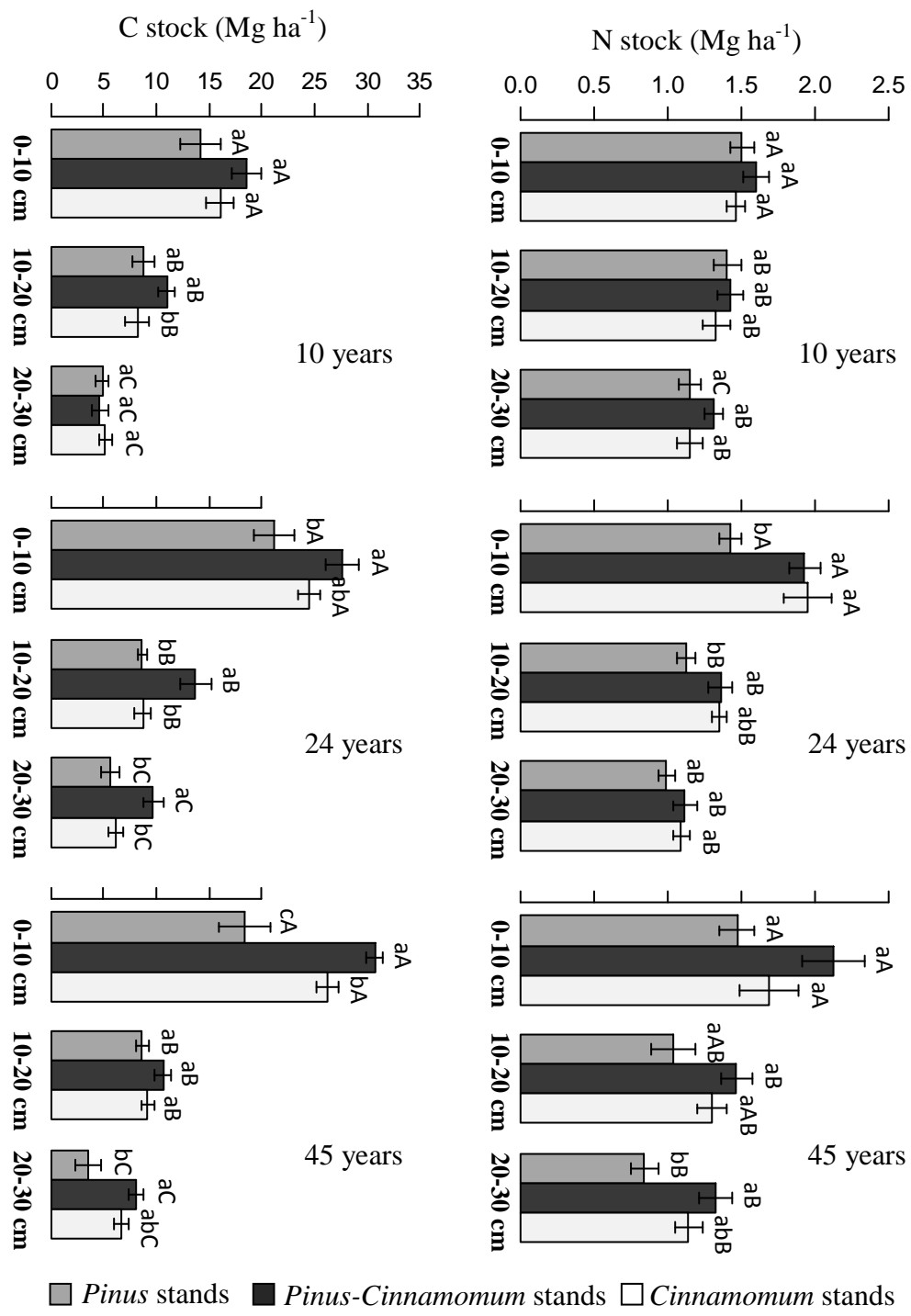

C stock (Mg ha$^{-1}$)

N stock (Mg ha$^{-1}$)

10 years

10 years

24 years

24 years

45 years

45 years

☐ *Pinus* stands  ■ *Pinus-Cinnamomum* stands  ☐ *Cinnamomum* stands

Fig. 4

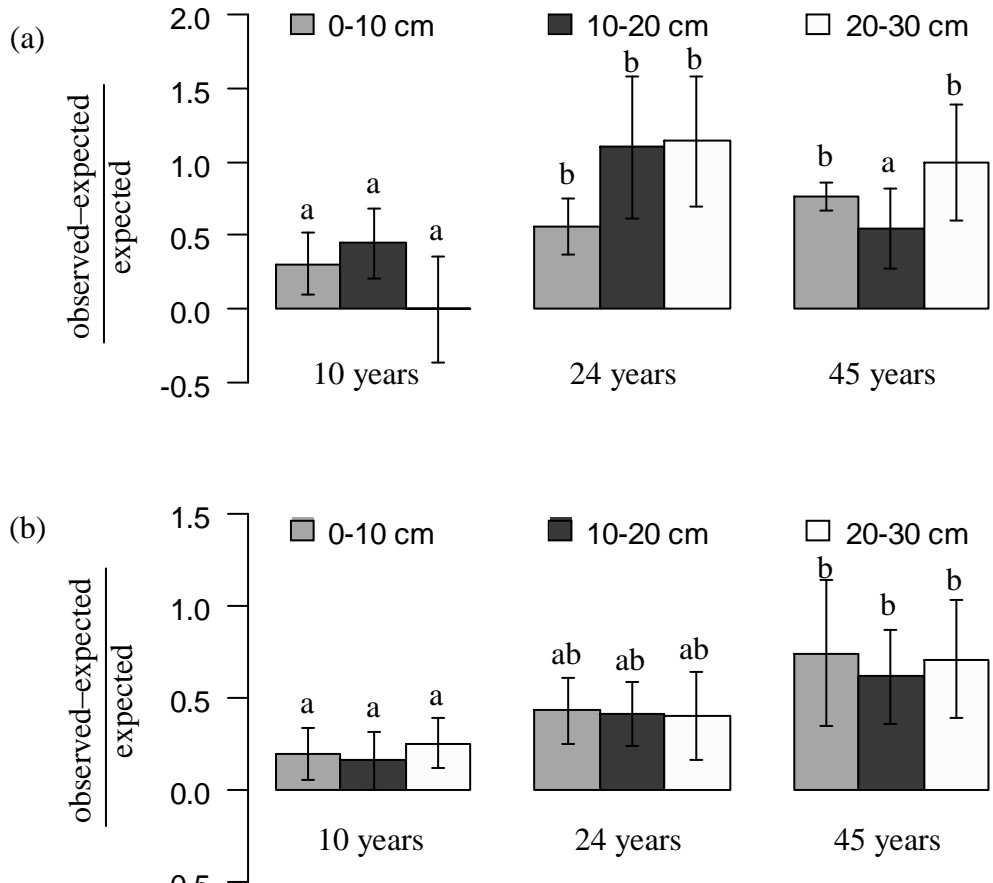