# Peer review of "Accumulation of soil organic C and N in planted forests fostered by tree species mixture"

_Biogeosciences, 2017_

## Referee Comment (RC1) · Anonymous Referee #1 · 11 May 2017

This manuscript shows the effects of mixed forests on the soil organic carbon and nitrogen stocks. This paper underline the benefits of mixed forests on the ecosystem functioning, and more precisely on a belowground compartment, which is largely underestimate. Currently, ecosystem services by soil, like carbon sequestration, are a hot topic and this paper bring some interesting results. The results are novel and very relevant for the future. I think the paper requires minor revision.

This study pointed the need to more information about the effect of mixed forest on soil C storage. The message is clear and concise and the manuscript is overall well written. The methods and statistical analyses seem appropriate. However, there are several critical issues should be further justified and/or addressed.

General comments: - About the chronosequence. In the materials and methods, your

description is too short. We need more information about the soil type and soil management in different station. The conditions required for a site to be included in the chronosequence study were that, with the exception of time, all soil-forming factors had remained constant since forest establishment. Could you add a table with all these soil informations? - About deep soil. Could you justify the soil depth of your experimentation? Some recent papers underline that the carbon could be sequestered in a deeper soil layer, more than 50 cm, and in a very stable form: humified. In page 2 line22, you didn't show any results of studies with soil deeper than 20 cm. - About roots. Page 7 line 9, you wrote "..in the same site (data not published) also suggested fine roots. ....". I think it will be a clear advance of this study to publish your data of roots here. With these data, you could confirm your pattern and highlight the distribution of carbon in the vertical and temporal ways. Roots data is quite difficult to obtain and it's a very relevant information. - What about the effect of more species? Could you add perspectives about that? And also the time effect after 45-yr ?

Specific comments: - P2, L11. Yes, but could you add the recent paper of Grossiord et al 2014 "Tree diversity does not always improve resistance of forest ecosystems to drought" doi: 10.1073/pnas.1411970111 - P4, L13-15. It's not clear. - P7, L7. Which soil depth ? - P7, L17. Need more information about "intraspecific competition". Competition for light and/or water? Do you have some results about that? There are some harvesting in pure stands during the chronosequence in order to decrease this competition? Roots data, could be nice also in this context. - Table 1. Could you add +/- SE (standard errors) in each value? - Table 2. P value = 0.0000, it's not realistic. I prefer P value < 0.0001 - Fig 3. Could you analyze also a differences among soil layers within the same stand? - Fig 4. Could you add a letters for significant differences?

---

## Author Comment (AC1) · 17 May 2017

Dear editor and referee:

Thank very much for the comments and suggestions, which is quite helpful to improve our manuscript. The soil conditions remained constant without disturbance after planting. The soil types at two stations were similar, but there were differences between them before the forest established. As the field in Taolin forestry station was used for nursery. We used this site just due to the scarcity of young plantations in Hunan Botanic Garden. More importantly, our purpose here is to evaluate the admixing effects on soil OC and N stocks by comparing mixed forests and corresponding monocultures over time, not to estimate the soil carbon and stock in mineral soil, which is also quite important of course. Therefore this field site was also used. This is also related the questions

you asked about the carbon sequestration in deeper soil layers, as we thought that the soil OC and N stock of shallow soil layer is most sensitive to tree species converting because of litter input by foliage and fine root turn over (Wang et al., 2013; Cremer et al., 2016). Of course, it is also quite intereting to know the C sequestration in the deeper soils. Regarding the fine root data, we are developing a manuscript for that. Here we only compared the two species mixtures with corresponding monocultures. It is quite interesting to know the diversity effects on soil C and N stocks when containing more species. But to find this kind of the field sites for this purpose are really difficult, even more difficult to find if the time effects were also considered. It is quite good suggestion adding some sentences about the perspectives of diversity effects containing more species and time effects after 45-yr.

Responses to the specific comments: P2, L11. Yes, but could you add the recent paper of Grossiord et al 2014 "Tree diversity does not always improve resistance of forest ecosystems to drought" doi: 10.1073/pnas.1411970111 Re: We will add that later.

P4, L13-15. It's not clear Re: Re: Here we only calculated the additive or non-additive effects of C and N stocks in mixed plantation by comparing the measured C or N stock with expected value, which was calculated from C or N stock in monocultures. To justify the additive or non-additive effects, we applied the method suggested by Ball et al. (2008) "Ball, B. A., Hunter, M. D., Kominoski, J. S., Swan, C. M., and Bradford, M. A.: Consequences of Non-Random Species Loss for Decomposition Dynamics: Experimental Evidence for Additive and Non-Additive Effects, J. Ecol., 96, 303–313, 2008." We will clarify them on the manuscript later.

P7, L7. Which soil depth ? Re: Her we divided the soil profiles to three layers (0-10cm, 10-20cm and 20-30cm). Here it refers the layer of 20-30 cm. we will clarify later in our manuscript.

P7, L17. Need more information about "intraspecific competition". Competition for light

and/or water? Do you have some results about that? There are some harvesting in pure stands during the chronosequence in order to decrease this competition? Roots data, could be nice also in this context. Re: We do not have specific data for that, so far. As there was no harvesting or thinning here, we think this is the case. Maybe we should add some references here.

Table 1. Could you add +/- SE (standard errors) in each value? Re: this can be done later

Table 2. P value = 0.0000, it's not realistic. I prefer P value < 0.0001 - Fig 3 Re: You are right, we will correct it in the manuscript.

Fig 3. Could you analyze also differences among soil layers within the same stand? Re: we can add the significant differences information here.

Fig 4. Could you add letters for significant differences? Re: Here our purpose was to see the magnitude trend of admixing effect over time. It can be done if it is necessary.

regards, Pifeng Lei On behalf of co-authors

―――――――――――――――

---

## Referee Comment (RC2) · Anonymous Referee #2 · 4 Jun 2017

This manuscript reports a study on SOC and N stocks in pure and mixed stands of Pinus and Cinnamomum at three different stand ages representing a chronosequence in the Hunan Province in China.

The aim and focus of the study are well justified, particular the idea to include the stand age effect in the assessment of tree species mixture effects. Several studies have previously studied mixture effects in young plantations (<20-30 years), so this is an interesting aspect of this study. I also appreciated the effort to address whether results were non-additive or just additive.

However, I have several concerns regarding this manuscript which prevents me from recommending it for publication. The experimental design, methods and the statistical analyses include several problematic assumptions and descriptions of methods are not

clear. This is a shame as the original idea of the study is relevant and justified.

The lack of information generally prevents me from being able to fully assess the work done. I have indicated that the manuscript could be reconsidered after major revisions, but this depends on whether the experimental design can support statistical analysis (see details below). If the authors can substantiate that their experimental design is appropriate for scientific evaluation or discuss this in a qualified manner, the revisions would entail a completely reworked statistical analysis as well as a description of sites and methods which fully matches the expectations of a transparent scientific paper. Only after this has been clarified can further details of the study be evaluated.

The experimental design compares stand composition effects with one site being 200 km away from the other two and provides not justification how we can assume that the main difference in SOC and n can be attributed to stand age and not the difference in site conditions. Mean climate conditions may be the same but soils are never the same as well as aspect, slope etc. Moreover I did not find any information about previous land use in these sites. The authors discuss that there are significant trends in SOC development with time depending on former land use, so this is very pertinent information to include and discuss. We would need much more detail about all these site conditions to be convinced that the two sites are "similar".

I have problems to understand the experimental design at each of the three sites, but it is relatively clear to me that there cannot be any proper replication of stand age as there is just one site per stand age. It is also unclear whether there were three subplots in each stand which were used as replicates in the statistical analysis – or if the individual cores were used as replicates in the analysis? In any event the statistical analyses must be flawed as we cannot separate the stand age effect from the site effect. Even within the site these subsamples within stands of a certain age are not real replicates for a statistical analysis.

Lastly the authors performed a three-way analysis of variance with all possible inter-

actions. This indicates use of pseudoreplicates, but apart from this they also include "depth" as a factor. This is highly problematic as the three layers are not independent. For instance, if a 0-10 cm layer has a high C concentration then the 10-20 cm layer and 20-30 cm layers underneath are highly likely to also have higher C concentrations. If depth is included in the statistical model, the authors need to account for this correlative structure in their data. This is similar to accounting for e.g. repeated measures analysis when analyzing a time series of data.

The most recent literature is not well referenced and addressed in the Introduction and the Discussion. For instance, Guckland et al. (2009) studied beech dilution gradients (J. Plant Nutr. Soil Sci. 172: 500-511) and more recently, Dawud et al. (2016, 2017) studied effects of tree species diversity gradients on soil C and N stocks in mature European stands (Dawud et al. 2016, Ecosystems 19: 645–660 ; Dawud et al. 2017, Funct. Ecol. 31: 1153-1162).

The data basis is a bit thin (only C and N concentrations and stocks in mineral soil), and the authors could leave out C and N concetrations from the main manuscript with no major loss of information. Instead, the paper would have been much stronger with inclusion of forest floor C and N stocks. Recent literature has shown that there are also clear dynamics in forest floor C and N stocks as a result of tree species mixtures. In addition the authors also mention a previous study of litterfall in the same sites, and unpublished root biomass data. I strongly suggest these data be included and discussed for a more coherent and strong publication (if the manuscript can be reworked for publication based on revision of the above-mentioned flaws).

The language of the manuscript needs substantial linguistic checking by a native English speaker. The wording is not correct in many places and several sentences are hard to understand.

---

## Author Comment (AC2) · 6 Jun 2017

Dear editor and anonymous referee: Thanks for your interest and critical comments on our MS. Please find responses below to respective comments. We hope the clarification below will get some positive feedbacks from you. regards, Pifeng Lei On behalf of co-authors

Responses to critical comments: The experimental design compares stand composition effects with one site being 200 km away from the other two and provides not justification how we can assume that the main difference in SOC and n can be attributed to stand age and not the difference in site conditions. Mean climate conditions may be the same but soils are never the same as well as aspect, slope

etc. Moreover I did not find any information about previous land use in these sites. The authors discuss that there are significant trends in SOC development with time depending on former land use, so this is very pertinent information to include and discuss. We would need much more detail about all these site conditions to be convinced that the two sites are "similar". RE: You are right about this experimental design. It would be nice to do this experiment in one site. However, we used two sites here due to the scarcity of young plantations in Hunan Botanic Garden. We also need to point out that our purpose in this study is to evaluate the admixing effects on soil OC and N stocks by comparing mixed forests and corresponding monocultures over time, not to compare the SOC and N stock along forest development in each forest types. Therefore, we selected three paired plantations, consisting of pure Pinus, Cinamonum plantation and Pinus-Cinamonum mixed plantation within the same age in one site. The principle of selection field site we used was to put three plantation types at the same age in one site. We stick to this principle as well during our analysis. Accordingly, we compared the magnitude of the admixing effects on the SOC and N over by using the relative values (e.g. percentages of over-performance in mixtures over monocultures, (observed-expected)/expected) at different stand ages, instead to comparing the absolute SOC and N stocks in one specific forest type with stand development. Therefore, the assumption of "the main difference in SOC and n can be attributed to stand age and not the difference in site conditions" would not happen here. In his point, we should justify it in our MS and clarify by adding this principle of site selection into our experimental design sections. For that we appreciate. And we also need to be more careful when discuss the SOC and N along chorosequence in discussion section. I have problems to understand the experimental design at each of the three sites, but it is relatively clear to me that there cannot be any proper replication of stand age as there is just one site per stand age. It is also unclear whether there were three subplots in each stand which were used as replicates in the statistical analysis – or if the individual cores were used as replicates in the analysis? In any event the statistical analyses must be flawed as we cannot separate the stand age

effect from the site effect. Even within the site these subsamples within stands of a certain age are not real replicates for a statistical analysis. RE: In this part, we must apologize for the confusion. As we have one previous publication (Wen et al. 2014) as reference, we did not make it clear in this MS. We select three stands, including pure Pinus, Cinamonum and Pinus-Cinamonum mixed plantations in each development stage (10, 24 and 45 years old). And three plots were established in each forest types at each development stage. Thereby our study included 27 plots consisting of Pinus-Cinnamomum mixed plantations and corresponding monocultures at age of 10, 24 and 45 years old. In each plot, 4 soil cores were sampled, sliced to three layers and treated as individual sample for SOC and N analysis. I hope this explanation is clearer and sound to you. Lastly the authors performed a three-way analysis of variance with all possible interactions. This indicates use of pseudoreplicates, but apart from this they also include "depth" as a factor. This is highly problematic as the three layers are not independent. For instance, if a 0-10 cm layer has a high C concentration then the 10-20 cm layer and 20-30 cm layers underneath are highly likely to also have higher C concentrations. If depth is included in the statistical model, the authors need to account for this correlative structure in their data. This is similar to accounting for e.g. repeated measures analysis when analyzing a time series of data. RE: The referee is right that we treated all the measurement as pseudoreplicates here simply to detect the effect of forest types, age and depth on SOC and N. We do need to improve this part. The most recent literature is not well referenced and addressed in the Introduction and the Discussion. For instance, Guckland et al. (2009) studied beech dilution gradients (J. Plant Nutr. Soil Sci. 172: 500-511) and more recently, Dawud et al. (2016, 2017) studied effects of tree species diversity gradients on soil C and N stocks in mature European stands (Dawud et al. 2016, Ecosystems 19: 645–660 ; Dawud et al. 2017, Funct. Ecol. 31: 1153-1162). RE: They are very good references, we would cite and combine them in our later revised version. The data basis is a bit thin (only C and N concentrations and stocks in mineral soil), and the authors could leave out C and N concetrations from the main manuscript with no major loss of information. Instead, the

paper would have been much stronger with inclusion of forest floor C and N stocks. Recent literature has shown that there are also clear dynamics in forest floor C and N stocks as a result of tree species mixtures. In addition the authors also mention a previous study of litterfall in the same sites, and unpublished root biomass data. I strongly suggest these data be included and discussed for a more coherent and strong publication (if the manuscript can be reworked for publication based on revision of the above-mentioned flaws). RE: It would be more informative to have more data, for example, the forest floor C and N stocks, although it is quite in these site that we used here. It is also typical in our area in subtropical area. Regarding the litterfall data, it was measured for other experiment and the data is not complete if we use it for our purpose as they only measured these three forests of 24 year old, instead of all the three age classes. For the fine root data, we recently developed one MS and submitted it Biogeosciences as a companion paper recently as encouraged by the previous anonymous reviewer. Hereby you could find it as supplement attachment. I hope you and the editor find it interesting. The language of the manuscript needs substantial linguistic checking by a native English speaker. The wording is not correct in many places and several sentences are hard to understand. RE: This can be done. We can improve it by correcting or send it for proof-reading.

Please also note the supplement to this comment:
http://www.biogeosciences-discuss.net/bg-2017-62/bg-2017-62-AC2-supplement.pdf

**Supplement:**

**Temporal changes of fine root overyielding and foraging strategies in planted monoculture and mixed forests**

Weiwei Shu[1, **], Xiaoxiao Shen[1, **], Pifeng Lei[1,2,*], Wenhua Xiang[1,2], Wende Yan[1,2]

[1] Faculty of Life Science and Technology, Central South University of Forestry and Technology, Changsha 410004, Hunan, China

[2] National Engineering Laboratory for Applied Technology of Forestry & Ecology in South China, Central South University of Forestry and Technology, Changsha 410004, Hunan, China

** *Authors were equally contributed*

* *Correspondence to:* Pifeng Lei (pifeng.lei@outlook.com)

**Abstract:** Mixed forests are believed to enhance ecosystem functioning and sustainability due to complementary resource use, environmental benefits and improved soil properties. The facilitation between different species may induce overyielding. Meanwhile, the species-specific fine root foraging strategies and tradeoff would determine the structure and dynamics of plant communities. Here the fine-root biomass, vertical distribution and morphology were investigated in *Pinus massoniana* and *Cinnamomum campho*ra pure and mixed plantations at 10-yr, 24-yr and 45-yr old stands. The results showed that the fine root biomass in the *Pinus -Cinnamomum* mixed forest exerted a certain degree of overyielding effect. These positive admixing effects, however, did not enhance with forest stand development. Instead, the magnitude of fine root overyielding in mixed forests showed a high degree of consistency with fine root biomass itself, suggesting the overyielding effects in mixed forests were correlated with the degree of belowground interaction and competition degree involved. The overall relative yield total (RYT) ranged from 1.83, and 1.51 to 1.33 in 10-yr-old, 24-yr-old and 45-yr-old stand, respectively. The overyielding were mainly attributed to the over-performance of conservative species, *Cinnamomum,* in mixed stands, as *Cinnamomum* accounted to the total fine root biomass 81.2%, 81.3% and 53.2% in 10, 24 and 45-yr-old *Pinus–Cinnamomum* mixed stands. In contrast, the pioneer species, *Pinus,* adapted to the presence of the species *Cinnamomum* by modification of vertical distribution and root morphological plasticity in the mixtures. The vertical fine root biomass distribution model showed *Pinus* roots shifted to the superficial layer when mixed with *Cinnamomum.* Furthermore, the specific root length (SRL) were significantly higher in *Pinus–Cinnamomum* mixed stands than that in *Pinus* monocultures, and the magnitude of differences increased over time. However, the vertical fine-root distribution and SRL for *Cinnamomum* did not show significant differences between monoculture and mixtures. These species-specific fine root foraging strategies might imply the differences of forest growth strategies of co-occurring species and contribute to the success and failure of particular species during the succession over time.

**1 Introduction**

In the global carbon cycle, roots of forest trees are an important reservoir of carbon, which is an important component of C pool in terrestrial ecosystem and plays a vital role on global carbon flux and carbon library (Vogt et al., 1998; Claus and George, 2011). In this context, in the past few decades, a lot of interests have been arosed with fine root biomass and production in forest, since fine roots accounted for as much as one-third of global net primary productivity (Jackson and Schulze, 1997) and are primary responsible for water and nutrient uptake by trees (Wells and Eissenstat, 2001;Zeng et al., 2015).

Belowground interactions among co-occurring species play critical roles on the community structure and distribution of plants. The plants are even capable of recognizing non-self neighbours and tend to proliferate more roots into substrate shared with coexisting species, and likely resulting in rooting aggregation and overyielding (de Kroon, 2007). Mixed forests are considered to be less susceptible to abiotic hazards like wind throw and snow break than pure stands (Schmid and Kazda, 2002; Reyer et al., 2010). Most studies investigating tree species diversity effects on aboveground and belowground productivity in forests, however, were based on comparisons between two species mixtures with monocultures. For example, a number of studies found a higher fine root biomass and production in mixtures (Schmid, 2002; Meinen et al., 2009b; Brassard et al., 2011; Lei et al., 2012), although other studies showed otherwise (Bolte and Villanueva, 2006; Mainen et al., 2009a). Recently, more studies were carried out to disentangle the effects of species diversity on the fine root biomass and production in mixed forests containing four to five tree species and the results were still equivocal (Rewald et al., 2009; Lei et al., 2012). However, all these studies on the relationship between diversity and productivity were based on one particular growth stage, or one static stand age. To our knowledge, recently only one study has been conducted comparing the effects on tree species on fine root productivity at 8 and 34 years old (Ma and Chen, 2016). Data on how fine root biomass, spatial distribution and morphology change in relation to stand age at pure and mixed forests, are comparatively few.

Fine root proliferation is greatly determined by environment conditions, such as nutrient supply in the soil, temperature and water. For example, Fine root vertical distribution is impacted by the spatial distribution of soil nutrients and moisture (Zhou and Shangguan, 2007), as well as soil structure and bulk density (Schenk, 2004). Besides, with increasing forests development, the proportion of fine root biomass was prone to increase in the top soil (Bouillet et al., 2002) or indicate no change (Claus and George, 2011). Furthermore, competition among individuals of the same species (intraspecific competition), as well as among different species populations (inter-specific competition), affects the process of tree root growth. Generally belowground competition depends on the soil exploitation capacity and exploitation efficiency of the fine-root systems of each plant, which were determined by the fine root biomass, surface area, root distribution within the soil horizons and specific root length (SRL) ( Bauhus et al., 2000; Makkonen and Helmisaari, 2001). Different plants within the community, in order to minimize competition for soil nutrients and moisture, may adjust the C investment to fine

roots and distribution, and/or morphological traits to adapt to the competition. Previous studies showed that beech developed a more dynamic and adaptive fine root foraging strategies, including biomass and vertical distribution, comparing to competitive species in mixed stands (Curt and Prevosto, 2003; Bolte et al., 2004). However, how the belowground interactions may shift with forest development is not clear, which may mirror the aboveground competition.

Here the temporal changes of fine root biomass, vertical distribution and fine root morphology were investigated in *Pinus massoniana* and *Cinnamomum camphora* mixed plantations and corresponding single species plantations at age of 10 years, 24 years and 45 years at 0-10 cm, 10-20 cm and 20-30 cm soil depth. Our objectives were to determine the magnitude of admixing effects on fine root biomass over time and to assess the possible shifts of foraging strategies for pioneer species (*P. massoniana*) and conservative species (*C. camphora*) along forest development. In this study we specifically tested the hypotheses that: (i) the total standing fine root biomass are higher in the mixed stands than those in corresponding monocultures, and the magnitude of positive admixing effect increases with forest development; (ii) the fine root foraging strategies of co-existing species, including fine root biomass, vertical distribution and morphological traits, in mixed forests mirrors the growth strategies of different species with forest development.

**2 Materials and methods**

**2.1 Field sites and experimental design**

This study was carried out in two different sites in Hunan province, China. One area is located in the Botanical Garden in Changsha (28°06'N, 113°02'E). The annual rainfall on this site is 1422 mm and the mean annual temperature is 17.2 °C, belonging to typical subtropical monsoon climate. The altitude ranges from 50 m to 100 m. The soil type is Alliti-Udic Ferrosols with well-drained clay-loam red soil developed from slate parent rock, and total N concentrations ranging from o.57 g kg$^{-1}$ to 1.56 g kg$^{-1}$ in top 30 cm depth of the soil profile (Wen et al., 2014). Single species and two species mixed patches, consisting of 24-year and 45-year old P. *massoniana* were selected. In monocultures and mixed stand, few *Pinus Elliotii* were also admixd here. Considering the similarity of growth characteristics and the difficulty of root separation between P. *massoniana* or P. *elliotii*, we treated them one group (thereafter called "*Pinus*"). Three plots of size 20m×20m were established in mixed *Pinus-Cinnamomum* stands at age of 24 and 45 years old and corresponding pure species stands (*Pinus* and *Cinnamomum*) at each age, amounting to 18 plots. The another site is located in Taolin forestry station (28°55'N,113°03'E) in Miluo county, approximate 200 km from the main site with similar climate and parent soil type. The mean annual precipitaion is about 1353.6 and mean annual temperature is 16.9 °C. Here, only the smaller patches of mixed and pure species stands with 10 years old pinus and cinnamomum were found. Therein three plots of 12m×12m in mixed forests stands and corresponding pure stands were set up as conducted above. Thereby our study consisted of 27 plots of mixed *Pinus–Cinnamomum* plantations and corresponding monocultures at age of 10, 24 and 45 years old. All the stems were recorded and selected site characteristics are presented in Table 1. More detailed information about the experimental site and soil condition referred to Wen et al. (2014).

**2.2 Fine root sampling and processing**

The root sampling was carried out in April 2013. Six soil cores in each plot were taken randomly in each square plot by using soil steel auger (diameter of 10 cm) to the soil depth of 30 cm and sliced to three layers (0-10, 10-20, and 20-30 cm). A preliminary survey had shown that very few fine roots occurred below 30 cm soil depth here. All the samples were labeled and transferred to plastic bag, sealed, and transported to the laboratory in 4℃ refrigerator.

In the laboratory, the processes of root separation off the soil were conducted with floatation method (Böhm, 1979; Lei et al., 2012a; 2012b). All the roots were collected with sieve of 0.65 mm aperture. The washed fine roots were poured and suspended in water, then sorted to *Pinus* and *Cinnamomum,* live and dead ones visually according to morphological traits, turgescence, root elasticity, colour, periderm surface structure, and exposure degree of steles. Living roots of *Pinus* and C. *Camphora* are intact, tough, and flexible. In contrast to that, dead roots were brittle and fractured easily and were distinguished by a dark to grey cortex and stele, or the complete loss of the stele and cortex. Only data on living fine roots (≤2.0mm in diameter) are reported in this study.

Live fine-root samples of each species were suspended in a water filled transparent tray on a scanner to facilitate samples dispersing. The morphological characteristics of fine roots were analyzed using the root scan analysis system WinRHIZO 2013 (Regent Instruments Inc., Quebec, Canada) by using images obtained. Thereafter, the root samples were oven-dried at 60 °C to constant weight. The specific root length (SRL) (m g$^{-1}$) was determined with the total root length by divided root dry weight.

**2.3 Data analysis**

All data were tested for a normal distribution with the Shapiro-Wilk test. Analysis of variance (ANOVA) or a non-parametric Mann–Whitney $U$ test was used to detect significant differences among three forest types. Differences between means were evaluated by Tukey's test of honestly significant difference. To examine whether overyielding occurred, the relative yield total (RYT) was calculated based on fine root biomass per basal area, as suggested by Lei (Lei et al., 2012b). The contributions of the different species, i.e. *Pinus and Cinnamomum,* to the relative yield total were calculated as the quotient of the fine-root biomass per basal area of each species at a particular stand age in the mixture to the counterpart value in the monoculture. RYT > 1 indicates overyielding, significant difference from 1 were analyzed using $t$ tests or the Mann–Whitney $U$ test.

To calculated the fine root vertical distribution, we adopted a commonly used equation developed by Gale and Grigal (1987):

$$Y = 1 - \beta^{d}$$

Therein, Y indicates the cumulative proportion of fine root biomass in the soil depth d (in cm). High values of β were indicate a large proportion of fine root at deeper soil depths, while low values indicate a large proportion of fine roots near the soil surface. Here we Used β as criterion to compare fine root vertical distribution of *Pinus* and *Cinnamomum* in *Pinus–Cinnamomum* mixed forests and corresponding monocultures as different ages. All data analyses were conducted with R (R 3.0.3, R development Core Team, Vienna, Austria).

**3 Results**

**3.1 Fine root biomass and overyielding**

As the forests grows, the standing fine root biomass tended to decrease with stand ages, averaged as 388.45 g m$^{-2}$, 269.27 g m$^{-2}$ and 138.59 g m$^{-2}$ in 30 cm soil depth in 10-yr-old, 24-yr-old and 45-yr-old stands, respectively. The standing fine root biomass was the highest in *Pinus–Cinnamomum* mixed stands compared to corresponding monocultures at 10- and 24-yr stands in 30 cm depth soil profile (Fig.1). In 45-yr-old stands, single *Pinus* stands showed highest fine root biomass than that in *Cinnamomum* stands and mixed *Pinus–Cinnamomum* stands, but significant differences were only detected in 10-yr-old forest stands. In mixed stands, *Cinnamomum* overperformed in fine root biomass contributions when compared with the aboveground abundance of mixed tree species. *Cinnamomum* accounted for the total fine root biomass 81.2%, 81.3% and 53.2% in 10, 24 and 45-yr-old *Pinus–Cinnamomum* mixed stands. Total fine root necromass showed lower values than fine root biomass and ranged from 17.84 g m$^{-2}$ to 96.54 g m$^{-2}$ in the pure *Pinus* ,*Cinnamomum* and mixed *Pinus–Cinnamomum* stands at differ ages (Fig.1). The standing fine root necromass was the highest in the pure *Cinnamomum* stands at 10- and 45-yr stands compared with mixed stands at the corresponding age in 30 cm depth soil profile, which differed significantly from each stands (p < 0.05). In 24-yr-old stands, however, the mixed stands showed the highest fine root necromass, although differences were not significant (p > 0.05).

Relative yield total (RYT) for each species in mixed plantations were calculated with adjusted fine root biomass per basal area. The results showed RYT of *Cinnamomum* were bigger than one in mixed plantations in all stand ages. Among them, only 45-yr-old forest stands showed an relative yield total value for fine root biomass that was not significantly different from one (p>0.05). For *Pinus,* the RYT showed inconsistent pattern that RYT was higher than one in 10-yr-old and 45-yr-old mixed stands, while the RYT was marginally lower than one in 24-yr-old stand (Fig. 2a). On the stand level, the overall RYT were bigger than one in all the three development stages, but the values of RYT seemed to decline with increasing stand ages, averaged as 1.83, 1.51 and 1.33 in 10-yr-old, 24-yr-old and 45-yr-old stand, respectively (Fig. 2b).

**3.2 Vertical fine root distribution**

The standing fine root biomass decreased gradually with soil depth, and fine root biomass was highest in 0-10cm for almost all the cases in the whole stand ages, which accounted for 56.0%, 51.7% and 47.2% in 10-yr-old, 24-yr-old and 45-yr-old stands. When compared with parallel forest stand within the same soil profiles, the fine root biomass was highest in the *Pinus–Cinnamomum* mixed stands in all the three soil layers in 10-yr-old stands and in the top soil layer in 24-yr-old stands. But the significant differences were detected only in 10-year-old stands (P<0.05) (Fig. 3). The abundance of species fine roots declined exponentially with increasing soil depth in the pure and mixed stands. Furthermore, the simulated rooting vertical distribution model of β value for for *Pinus* and *Cinnamomum* growing in pure stands at different stand ages showed the similar patterns that both species allocated more fine roots to the deeper layer with increasing stand ages. Compared the β values for *Pinus* and

*Cinnamomum* in the pure and mixed stands, however, the results showed that the adjusted β value For *Pinus* in pure stands were significantly higher than that in the mixed stands at all forest ages (see Fig 4), indicating fine roots of *Pinus* were more concentrated in the top soil when mixed with *Cinnamomum*. The β value was 0.915, 0.937, 0.939 in pure stands, and 0.911, 0.914, 0.925 in mixed

5    stands in forests age of 10-yr old, 24-yr old, and 45-yr old, respectively. For *Cinnamomum*, the β values did not show consistent pattern along the stand age as it increased in the pure stands, but decreased in the mixed stands along chronosequence.

**3.3 Specific root length**

The comparison of specific root length (SRL) in the pure and mixed stands revealed striking

10    differences for *Pinus* and *Cinnamomum*. The SRL*Pinus* in mixed stands were significantly higher than those of corresponding pure stand, and the differences becames more pronounced over time. For *Cinnamomum*, however, structural trait did not show regular pattern in pure and mixed stand plots. Result from one-way ANOVA revealed that both plantation type and stands age had significant effects on morphology. At in the pure *Pinus* stand, the SRL changed along the chronosequence, decreasing

15    from 8.84 m g$^{-1}$ in the 10-yr old stands to 6.72 m g$^{-1}$ in the 24-yr old stands and 6.29 m g$^{-1}$ in the 45–yr old stands. However, the SRL of mixed stands increased along chronosequence. And significant differences were only detected between the 24-yr and 45-yr old stands (p < 0.05) (Fig 5a). For species *Cinnamomum*, the SRL of *Cinnamomum* fine root in the pure stands and mixed stands was basically similar, ranging from 5.24 m g$^{-1}$ to 8.90 m g$^{-1}$, and its value did not significantly differ within the same

20    ages (p>0.05) (Fig. 5). Comparing with different age stages, the results showed that SRL of *Cinnamomum* was increased with increase stand age in the pure stands, but no significant differences were detected. However, this kind of phenomenon does not show in the *Pinus–Cinnamomum mixed* stands.

**4 Discussion**

25    In this study, the fine root biomass seemed to decrease in pure and mixed stands with increase stand age. The stand density decreased with development stages here. The high tree density may have accounted for the higher root biomass, which is in agreement with a previous report described for 13-year-old postfire lodgepole *pine* forests, where the fine root biomass increased with tree density (Creighton M Litton, 2003). When compared with three types of plantations, the fine root biomass was

30    higher in the *Pinus–Cinnamomum* mixed stands than those in the pure *Pinus* and *Cinnamomum* stands in 30 cm soil depth in 10-yr-old and 24-yr-old stands. Many previous studies reported the similar pattern that species-rich forests exhibited higher fine root biomass than species-poor stands (Brassard et al., 2011; Liu et al., 2011). Although the absolute fine root biomass in the *Pinus–Cinnamomum* mixed forests were not significantly higher than that the counterpart monocultures in 24-yr-old, and even

35    lower than that in monocultures in 45-yr-old stands, the RYT was higher than one, suggesting overyield when comparing the adjusted fine root biomass per basal area in mixed forest with monocuters (Fig. 2b) (Hector, 2006). Likewise, fine root overyielding was reported in mixed forests of *Eucalyptus grandis*

and *Acacia mangium* stands (Laclau et al., 2013), as well as in European beech, sessile oak, Norway spruce and Douglas fir mixed stands at two-,three-,and four-species neighbourhoods in comparison to monocultures (Lei et al., 2012b).

Here we primarily attempted to assess the variations of these admixing effects over time and expected that the admixing effect would be more pronounced over time, as the interactions between different species become more intense over time. On the contrary, in this study, the magnitude of over-yielding in the *Pinus–Cinnamomum* mixed forests declined with stand development as shown in Fig. 1 and Fig. 2. The significant differences between the *Pinus–Cinnamomum* mixed forests and corresponding monocultures were only detected in10-yr-old forests. Besides, the direct evidence showed that relative yield total decreased from 1.83, and 1.51 to 1.33 in 10-yr-old, 24-yr-old and 45-yr-old stand, respectively. This pattern is consistent the trend that the standing fine root biomass decreased with stand. Therefore, it is likely that the magnitude of fine root overyielding in mixed forests was correlated with fine root biomass and the belowground competitive degree involved.

Here we calculated the RYT for the component species in the mixtures to estimate the specific performance and dynamics of each species to the overyielding in the mixtures over time, and foraging strategies as well. In the mixtures, the RYT of conservative species, *Cinnamomum,* was higher than one for all the stand development stages and the differences were significant from one in 10-yr-old and 24-yr-old stands. This pattern was supported by data on fine root biomass, which *Cinnamomum* accounted for 81.2%, 81.3% and 53.2% in 10, 24 and 45-yr-old *Pinus–Cinnamomum* mixed stands, respectively. (Fig. 1), suggesting *Cinnamomum* invested more carbon to belowground fine roots when co-occurring species presents. In contrast, the pioneer species, *Pinus,* showed significant higher RYT only in 10-yr-old stand and then fluctuated from one thereafter.

In our study, we compared the vertical distribution of fine root biomass with the exponential model of Gale and Grigal (Gale and Grigal 1987) and found very similar patterns for the overall distribution in pure stands along chronosequence. Two species showed the exponential indicator (β values) increased with stand development. The β data indicated that there was clear spatial separation of the fine root systems of the *Pinus* within 0–30 cm of the soil profile. The *Pinus* roots occupied the deeper soil layers in the pure stand whereas it shifted to the superficial layers when mixed with *Cinnamomum*. Many researches revealed previously significant effect of mixed stand on fine root distributions. Bolte and Villanueva (Bolte and Villanueva, 2006) suggested that fine root of *P. abies* distributed deeper in mixed stands than pure stands. Moreover, in mixed stand of beech and *Quercus Petraea*, fine roots of beech grew more deeper than fine roots of *Quercus Petraea* (Büttner and Leuschner, 1994). The presence of *Cinnamomum* in the mixed stands could have pushed the fine root system of *pinus* towards the soil surface where the water and nutrient were more enriched. In mixed stand fine roots tend to proliferate and compete with neighbors for nutrients and water (Leuschner et al., 2000; Schenk and Jackson, 2005) by developing a more flexible fine root system when there is more intense belowground competition. However, the fine-root distribution data indicated that there was no obvious regularity spatial separation of the fine-root systems for *Cinnamomum* within 0-30 cm of the soil profile. Therefore, different tree species may have different strategies for the presence of neighbor species in terms of vertical niche separation.

The specific root length (SRL) was used as indicator for nutrient uptake efficiency and responses to environmental changes or competition (Ostonen et al., 2007). SRL can response that plants were in root

growth the efficiency of consumption photosynthetic primary product, high SRL indicate high efficiency of using photosynthetic primary product of plant root systems (Pregitzer et al., 1998). Our studies showed that SRL of *Pinus* decreased slightly with increase forest age in monocultures, but increased with stand age in mixtures. In 45-yr-old stand, the SRL of *Pinus* in the *Pinus-Cinnamomum* mixed stand was up to two-fold higher than that in monocultures, suggesting *Pinus* exploited water and nutrient resources more efficiently when growing admixed to *Cinnamomum*. It is likely that pioneer species, *Pinus*, was stressed by the competition from the later successional conservative species, *Cinnamomum.* The results were consistent with of previous study showing that compared to pure beech, the higher specific root length (SRL) and specific surface area (SSA) were found for beech admixed with spruce (Bolte and Villanueva 2006). For *Cinnamomum,* in contrast, fine root morphology was rather similar in pure and mixed stands. The SRL of *Cinnamomum* seemed to increase along chronosequence, but no significant differences were detected, in agreement of previous study that mean SRL was not significantly different among the beech, oak and alder chronosequences (Jagodzinski et al., 2016). The contrast performances between early and late successional species, for example, higher carbon input into fine root biomass for conservative species *Cinnamomum*, and shallower fine root distribution and higher SRL in mixtures for pioneer species *Pinus* may suggest that the rooting strategies for competition from co-occurring species was species-specific.

**Acknowledgements**

This study was sponsored by National Natural Science Foundation of China (31200346), and Introduce Talent Fund of CSUFT. We thank Forest Administration on Hunan and Forest Station of Taolin for the permission to use the site. Furthermore, we acknowledge the assistance of Fang Jiang, Yuqin Xu and Hao Yi in the field and laboratory.

**References**

Bauhus, J., Khanna, P.K., and Menden, N.: Aboveground and belowground interactions in mixed plantations of Eucalyptus globulus and Acacia mearnsii, Can. J. Forest Res., 30, 1886-1894, 2000.

Böhm, W.: Methods of studying root systems. Springer Verlag, Berlin, 1979.

Bolte, A., Rahmann, T., Kuhr, M., Pogoda, P., Murach, D., and Gadow, K. V.: Relationships between tree dimension and coarse root biomass in mixed stands of European beech (Fagus sylvatica L.) and Norway spruce (Picea abies[L.] Karst.), Plant Soil, 264, 1-11, 2004.

Bolte, A., and Villanueva, I.: Interspecific competition impacts on the morphology and distribution of fine roots in European beech (Fagus sylvatica L.) and Norway spruce ( Picea abies (L.) Karst.), Eur. J. Forest Res., 125, 15-26 , 2006.

Bouillet, J. P., Laclau, J. P., Arnaud, M., M'Bou, A. T., Saint-André, L., and Jourdan, C.: Changes with age in the spatial distribution of roots of Eucalyptus clone in Congo: Impact on water and nutrient uptake, Forest Ecol. Manag., 171, 43-57, 2002.

Brassard, B. W., Chen, H., Bergeron, Y., and Paré, D.: Differences in fine root productivity between mixed- and single-species stands, Funct. Ecol., 25, 238-246, 2011.

Büttner, V., and Leuschner, C.: Spatial and temporal patterns of fine root abundance in a mixed oak-beech forest, Forest Ecol. Manag., 70, 11-21, 1994.

5   Claus, A., and George, E.: Effect of stand age on fine-root biomass and biomass distribution in three European forest chronosequences, Can. J. Forest Res., 35, 1617-1625, 2011.

Curt, T., and Prevosto, B.: Root biomass and rooting profile of naturally regenerated beech in mid-elevation Scots pine woodlands, Plant Ecol., 167, 269-282, 2003.

de Kroon, H.: How do roots interact? Science 318, 1562-1563, 2007.

10  Gale, M. R., and Grigal, D. F.: Vertical root distributions of northern tree species in relation to successional status, Can. J. Forest Res., 17, 829-834, 1987.

Hector, A.: Overyielding and stable species coexistence, New Phytol., 172, 1–3, 2006.

Jackson, R. B., Mooney, H. A., and Schulze, E. D.: A Global Budget for Fine Root Biomass, Surface Area, and Nutrient Contents, Proc. Natl. Acad. Sci., 94, 7362-7366, 1997.

15  Jagodzinski, A. M., Ziółkowski, J., Warnkowska, A., and Prais, H.: Tree Age Effects on Fine Root Biomass and Morphology over Chronosequences of Fagus sylvatica, Quercus robur and Alnus glutinosa Stands, Plos One, 11, e0148668, 2016.

Laclau, J. P., and Bouillet, J. P.: Mixing eucalyptus and acacia trees leads to fine root over-yielding and vertical segregation between species, Oecologia, 172, 903-13, 2013.

20  Lei, P., Scherer-Lorenzen, M., and Bauhus, J.: Belowground facilitation and competition in young tree species mixtures, Forest Ecol. Manag., 265 , 191-200, 2012a.

Lei, P., Scherer-Lorenzen, M., and Bauhus, J.: The effect of tree species diversity on fine-root production in a young temperate forest, Oecologia, 169, 1105-1115, 2012b.

Leuschner, C., Hertel, D., Coners, H. and Büttner, V.: Root competition between beech and oak: a
25      hypothesis, Oecologia, 126, 276-284, 2000.

Leuschner, C., Hertel, D., Schmid, I., Koch, O., Muhs, A., and Hölscher, D.: Stand fine root biomass and fine root morphology in old-growth beech forests as a function of precipitation and soil fertility, Plant Soil, 258, 43-56, 2004.

Litton, C. M., Ryan, M. G., Tinker, D. B., and Knight, D. H.: Belowground and aboveground biomass
30      in young postfire lodgepole pine f, Can. J. Forest Res., 33, 351-363, 2003.

Liu, C., Xiang, W.H., Tian, D.L., Fang, X., and Peng, C.H.: Overyielding of fine root biomass as increasing plant species richness in subtropical forests in central southern china, Acta Phytoecol. Sinica, 35, 539-550, 2011(in Chinese with English abstract).

Ma, Z., and Chen, H. Y. H.: Effects of species diversity on fine root productivity increase with stand
35      development and associated mechanisms in a boreal forest, Journal of Ecology., doi: 10.1111/1365-2745.12667, 2017.

Makkonen, K., and Helmisaari, H. S.: Fine root biomass and production in Scots pine stands in relation to stand age, Tree Physiol., 21, S114, 2001.

Meinen, C., Hertel, D., Leuschner, C.: Biomass and morphology of fine roots in temperate broad-leaved forests differing in tree species diversity: is there evidence of below-ground overyielding? Oecologia, 161, 99-111, 2009a.

Meinen, C., Leuschner, C., Ryan, N. T., and Hertel, D.: No evidence of spatial root system segregation and elevated fine root biomass in multi-species temperate broad-leaved forests, Trees, 23, 941-950. 2009b.

Ostonen, I., Lõhmus, K., Helmisaari, H. S., Truu, J., and Meel, S.: Fine root morphological adaptations in scots pine, norway spruce and silver birch along a latitudinal gradient in boreal forests, Tree Physiol., 27, 1627, 2007.

Ostonen, I., Püttsepp, Ü., Biel, C., Alberton, O., Bakker, M.R., Lohmus, K., Majdi, H., Metcalfe, D., Olsthoorn, A.F.M., Pronk, A., Vanguelova, E., Weith, M., and Brunner, I.: Specific root length as an indicator of environmental change. Plant Biosystems, 141, 426-442, 2007.

Pregitzer, K.S., Laskowski, M.J., Burton, A.J., Lessard, V.C. and Zak, D.R.: Variation in sugar maple root respiration with root diameter and soil depth, Tree Physiol., 18, 665-670, 1998.

Rewald, B., and Leuschner, C.: Belowground competition in a broad-leaved temperate mixed forest: pattern analysis and experiments in a four-species stand, Can. J. Forest Res., 128, 387-398, 2009.

Reyer C., Lasch P., Mohren G.M.J., and Sterck F.J.: Inter-specific competition in mixed forests of Douglas-fir (*Pseudotsuga menziesii*) and common beech (*Fagus sylvatica*) under climate change – a model-based analysis. Ann. For. Sci., 67, 805, 2010

Schenk, H. J.: Progress in Botany Vertical Vegetation Structure Below Ground: Scaling from Root to Globe, 341-373 , 2004.

Schenk, H. J. and Jackson, R. B.: Mapping the global distribution of deep roots in relation to climate and soil characteristics, Geoderma, 126, 129-140, 2005.

Schmid, I.: The influence of soil type and interspecific competition on the fine root system of Norway spruce and European beech, Basic. Appl. Ecol., 3, 339-346, 2002.

Schmid, I., and Kazda, M.: Root distribution of Norway spruce in monospecific and mixed stands on different soils, Forest Ecol. Manag., 159, 37-47, 2002.

Vogt, K. A., Vogt, D. J., and Bloomfield, J.: Analysis of some direct and indirect methods for estimating root biomass and production of forests at an ecosystem level, Plant Soil, 200, 71-89, 1998.

Wells, C. E., and Eissenstat, D. M.: Marked Differences in Survivorship among Apple Roots of Different Diameters, Ecology, 82, 882-892, 2001.

Wen, L., Lei, P., Xiang, W., Yan, W., and Liu, S.: Soil microbial biomass carbon and nitrogen in pure and mixed stands of Pinus massoniana and Cinnamomum camphora in stand age, Forest Ecol. Manag., 328, 150-158, 2014.

Zeng, W., Zhou, B., Lei, P., Zeng, Y., Liu, Y., and Liu, C., et al.: A molecular method to identify species of fine roots and to predict the proportion of a species in mixed samples in subtropical forests, Front. Plant Sci., 6, 313, 2015.

Zhou, Z., and Shangguan, Z.: Vertical distribution of fine roots in relation to soil factors in Pinus tabulaeformis Carr. forest of the Loess Plateau of China, Plant Soil, 291, 119-129, 2007.

**Table 1** Stand characteristics in pure species *Pinus* stands, pure *Cinnamomum* stands and mixed *Pinus-Cinnamomum* stands at the age 10, 24 and 45 years old.

| stand | age | species | Density (n ha$^{-1}$) | diameter at breast height (cm) | Height (m) | Basal area (m$^2$ ha$^{-1}$) |
|---|---|---|---|---|---|---|
| *Pinus stands* | 10 | *Pinus* | 2592 | 9.38 | 5.28 | 20.06 |
| | 24 | *Pinus* | 2050 | 14.18 | 12.86 | 35.37 |
| | 45 | *Pinus* | 600 | 21.40 | 12.47 | 22.84 |
| *Cinnamomum stands* | 10 | *Cinnamomum* | 2708 | 7.77 | 5.99 | 14.26 |
| | 24 | *Cinnamomum* | 900 | 17.02 | 13.71 | 23.46 |
| | 45 | *Cinnamomum* | 800 | 21.06 | 13.24 | 30.63 |
| Mixed *Pinus-Cinnamomum* stands | 10 | *Pinus* | 902 | 7.64 | 4.73 | 4.37 |
| | | *Cinnamomum* | 1689 | 8.14 | 7.20 | 9.83 |
| | 24 | *Pinus* | 267 | 18.88 | 12.35 | 7.80 |
| | | *Cinnamomum* | 592 | 15.27 | 11.41 | 12.45 |
| | 45 | *Pinus* | 250 | 19.69 | 12.37 | 7.91 |
| | | *Cinnamomum* | 325 | 21.94 | 13.75 | 12.91 |

**Figure captions**

**Fig. 1** Live fine root biomass and fine root necromass in pure *Pinus* stand (P), *Cinnamomum* stand (C) and mixed *Pinus-Cinnamomum* stand (PC) in 0-30 cm soil depth at the age of 10, 24 and 45 years. Error bars indicate standard errors. Different letters indicate significant differences among different stands within the same ages (p<0.05).

**Fig.2** *Relatvie yield total for each species (a) and* Relative yield total for total standing fine root biomass *(b) and in Pinus-Cinnamomum mixed stands* at ages of 10, 24, and 45 years old stand age in comparison to monocultures (reference level = 1). *Asterisks* denote significant differences from 1 with t test or Mann–Whitney *U* test, *P* < 0.05. Each datum shows the mean ± SE.

**Fig. 3** Fine root biomass in pure *Pinus*, *Cinnamomum* and mixed *Pinus-Cinnamomum* stands in 0-10 cm, 10-20 cm and 20-30 cm soil depth at the age of 10, 24 and 45 years. Error bars indicate standard error. Different letters indicate significant differences among different stands within the same soil profile and age stages (*p*<0.05).

**Fig.4** Cumulative fine root biomass along the soil profiles and the coefficients of the rooting distribution ( β ) for *Pinus* (a) and *Cinnamomum* (b) in the pure and mixed stands at ages of 10, 24 and 45 years. The  β value indicates the degree of fine root biomass decreases with soil depth.

**Fig.5** Specific root length for live fine root in pure *Pinus, Cinnamomum* and *Pinus-Cinnamomum* mixed stands at the age of 10, 24 and 45 years. Error bars indicate standard errors. *Asterisks* indicates significant differences between pure and mixed stands for *P. massoniana* or *C. camphora* within the same age stages (p<0.05).

**Fig. 1**

[Figure]

**Fig. 2**

[Figure]

**Fig.3**

[Figure]

**Fine root biomass (g m⁻²)**

*Pinus* stands   *Pinus-Cinnamomum* stands   *Cinnamomum* stands

**Fig.4**

[Figure]

**Fig. 5**

[Figure]

---

## Author Response (AR1)

*Dear editor*

*Please find below our responses to the comments of the reviewers and explanations how we addressed the critical points. We are grateful for the very constructive suggestions, which certainly helped to improve our manuscript.*
*Below, you will find our responses in **italics** to the respective comments.*

*With best regards,*
*Pifeng Lei*
*On behalf co-authors*

**Responses to Reviewer # 1 ##**

This manuscript shows the effects of mixed forests on the soil organic carbon and nitrogen stocks. This paper underline the benefits of mixed forests on the ecosystem functioning, and more precisely on a belowground compartment, which is largely underestimate. Currently, ecosystem services by soil, like carbon sequestration, are a hot topic and this paper bring some interesting results. The results are novel and very relevant for the future. I think the paper requires minor revision.

This study pointed the need to more information about the effect of mixed forest on soil C storage. The message is clear and concise and the manuscript is overall well written. The methods and statistical analyses seem appropriate. However, there are several critical issues should be further justified and/or addressed.

General comments: - About the chronosequence. In the materials and methods, your description is too short. We need more information about the soil type and soil management in different station. The conditions required for a site to be included in the chronosequence study were that, with the exception of time, all soil-forming factors had remained constant since forest establishment. Could you add a table with all these soil informations? - About deep soil. Could you justify the soil depth of your experimentation? Some recent papers underline that the carbon could be sequestered

in a deeper soil layer, more than 50 cm, and in a very stable form: humified. In page 2 line22, you didn't show any results of studies with soil deeper than 20 cm. – About roots. Page 7 line 9, you wrote "..in the same site (data not published) also suggested fine roots.....". I think it will be a clear advance of this study to publish your data of roots here. With these data, you could confirm your pattern and highlight the distribution of carbon in the vertical and temporal ways. Roots data is quite difficult to obtain and it's a very relevant information. - What about the effect of more species? Could you add perspectives about that? And also the time effect after 45-yr ?

*RE: Thank very much for the comments and suggestions, which is quite helpful to improve our manuscript. The soil conditions remained constant without disturbance after planting owning to either the foundation of Botanic Garden or the young age of the forests. The soil types at two stations were similar, but there were differences between them before the forest establishment. As the field in Taolin forestry station was used for nursery and abandoned for three years before planting. We used this site just due to the scarcity of young plantations in Hunan Botanic Garden. More importantly, our purpose here is to evaluate the admixing effects on soil OC and N stocks by comparing mixed forests and corresponding monocultures over time, not to estimate the soil carbon and stock in mineral soil, which is also quite important of course. Therefore, what we considered was to make sure the three parallel forests at the same age are located in one site. See P3, L20-25.*

*Here we collected the soil samples down to 30 cm depth as the soils in this area were susceptible to the environmental variations and sensitive to the carbon input by litter and fine roots, which matches the purposes to assess the admixing effects on soil OC and N stock over time (Wang et al., 2013; Cremer et al., 2016). Of course, it is also quite interesting to know the C sequestration in the deeper soils. See P4, L1-3.*

*Regarding the fine root data, we have developed a manuscript for that and submitted to BMC Ecology recently. See P8, L1-3.*

*Here we only compared the two species mixtures with corresponding monocultures. It*

*is quite interesting to know the diversity effects on soil C and N stocks when containing more species. But to find this kind of the field sites for this purpose are really difficult, even more difficult to find when the time effects were also considered. It is quite good suggestion adding some sentences about the perspectives of diversity effects containing more species and time effects after 45-yr. See P8, L24-25.*

Responses to the specific comments:

P2, L11. Yes, but could you add the recent paper of Grossiord et al 2014 "Tree diversity does not always improve resistance of forest ecosystems to drought" doi: 10.1073/pnas.1411970111

*Re: added as suggested. See P2, L11.*

P4, L13-15. It's not clear

*Re: Here we rephrased as "Then these expected values of OC or N stock were compared with observed values for each individual sample that were measured experimentally in mixed stands with formula: (observed-expected)/expected. For soil OC and N stock in specific soil depth at given stand ages, 95% confidence intervals (CI) were calculated with above formula. If the CIs for mixtures did not cross y = 0, the admixing effect was considered non-additive (Ball et al., 2008)". See P4, L22-26*

P7, L7. Which soil depth?

*Re: Here, we deleted that sentence and replace with "In our results the soil OC and N stocks decreased significantly with increasing soil depth. However, the magnitude of over-performance in the Pinus-Cinnamomum mixed stands over monocultures increased with soil depth (Fig 3 and Fig. 4)", since the added analysis of significant differences in Fig.3 and Fig.4 as suggested makes the results clearer. See P7, L31-33.*

P7, L17. Need more information about "intraspecific competition". Competition for light and/or water? Do you have some results about that? There are some harvesting in pure stands during the chronosequence in order to decrease this competition? Roots

data, could be nice also in this context.

*Re: We do not have data for that, so far. As there was no harvesting or thinning for all the plantations here, we stated that as general phenomenon as the partitioning use of light and soil resources by different species, which induce overyielding and higher input of litter and fine roots to soil. We also added one reference here. see P2, L13. Regarding the roots data, please see P8, L1-3.*

Table 1. Could you add +/- SE (standard errors) in each value?

*Re: Here we added $\pm$ standard deviation.*

Table 2. P value = 0.0000, it's not realistic. I prefer P value < 0.0001 –

*Re: Corrected as suggested.*

Fig 3. Could you analyze also differences among soil layers within the same stand?

*Re: Analyzed and marked in the Figure as suggested.*

Fig 4. Could you add letters for significant differences?

*Re: Added as suggested. Here our purpose was to see the magnitude trend of admixing effect over time. So we added the significant differences among different ages within the same soil layers.*

Responses to Reviewer # 2 ##

(1) The experimental design compares stand composition effects with one site being 200 km away from the other two and provides not justification how we can assume that the main difference in SOC and n can be attributed to stand age and not the difference in site conditions. Mean climate conditions may be the same but soils are never the same as well as aspect, slope etc. Moreover I did not find any information about previous land use in these sites. The authors discuss that there are significant trends in SOC development with time depending on former land use, so this is very pertinent information to include and discuss. We would need much more detail about

all these site conditions to be convinced that the two sites are "similar".

*RE: You are right about this experimental design. It would be nice to do this experiment in one site. Here we used two sites due to the scarcity of young plantations in Hunan Botanic Garden. Here we do need to justify the site selection in the method section to point out specifically that our purpose in this study is to evaluate the admixing effects on soil OC and N stocks over time by comparing mixed forests and corresponding monocultures at three forest development stages, not to compare the absolute values of SOC and N stock along forest development in each forest types. For this purpose, we strictly selected three paired plantations within the same age in one site, consisting of pure Pinus, Cinamonum plantation and Pinus-Cinamonum mixed plantation. The principle of selection field site we used was to make sure three plantation types at the same age in one site. We stick to this principle as well during our analysis. Accordingly, we compared the magnitude of the admixing effects on the SOC and N over by using **the relative values (e.g. percentages of over-performance in mixtures over monocultures, (observed-expected)/expected)** at different stand ages, instead to comparing the absolute SOC and N stocks in one specific forest type with stand development. This is how we did to avoid the assumption of "the main difference in SOC and n can be attributed to stand age and not the difference in site conditions".*

(2) I have problems to understand the experimental design at each of the three sites, but it is relatively clear to me that there cannot be any proper replication of stand age as there is just one site per stand age. It is also unclear whether there were three subplots in each stand which were used as replicates in the statistical analysis – or if the individual cores were used as replicates in the analysis? In any event the statistical analyses must be flawed as we cannot separate the stand age effect from the site effect. Even within the site these subsamples within stands of a certain age are not real replicates for a statistical analysis.

*RE: In this part, we apologize for the confusion. As we have one previous publication (Wen et al. 2014) as reference, we did not make it clear in our previous MS. We*

*selected three stands, including pure Pinus, Cinamonum and Pinus-Cinamonum mixed plantations in each development stage (10, 24 and 45 years old). And three plots were established in each forest stands at each development stage as three replicates. Thereby this experiment included 27 plots consisting of Pinus-Cinnamomum mixed plantations and corresponding monocultures at age of 10, 24 and 45 years old. In each plot, 4 soil cores were sampled, sliced to three layers and treated each of them as individual sample for SOC and N analysis. I hope this explanation is clearer and sound. See P3, L7-14 and P3, L29-30.*

(3) Lastly the authors performed a three-way analysis of variance with all possible interactions. This indicates use of pseudoreplicates, but apart from this they also include "depth" as a factor. This is highly problematic as the three layers are not independent. For instance, if a 0-10 cm layer has a high C concentration then the 10-20 cm layer and 20-30 cm layers underneath are highly likely to also have higher C concentrations. If depth is included in the statistical model, the authors need to account for this correlative structure in their data. This is similar to accounting for e.g. repeated measures analysis when analyzing a time series of data.

*RE: Yes, Here we treated all the measurement as pseudoreplicates here simply to detect the effect of forest types, age and depth on SOC and N.*

(4) The most recent literature is not well referenced and addressed in the Introduction and the Discussion. For instance, Guckland et al. (2009) studied beech dilution gradients (J. Plant Nutr. Soil Sci. 172: 500-511) and more recently, Dawud et al. (2016, 2017) studied effects of tree species diversity gradients on soil C and N stocks in mature European stands (Dawud et al. 2016, Ecosystems 19: 645–660 ; Dawud et al. 2017, Funct. Ecol. 31: 1153-1162).

*RE: They are very good references, we cited and combined them in our revised version. See P6, L19-22.*

(5) The data basis is a bit thin (only C and N concentrations and stocks in mineral soil), and the authors could leave out C and N concentration from the main

manuscript with no major loss of information. Instead, the paper would have been much stronger with inclusion of forest floor C and N stocks. Recent literature has shown that there are also clear dynamics in forest floor C and N stocks as a result of tree species mixtures. In addition the authors also mention a previous study of litterfall in the same sites, and unpublished root biomass data. I strongly suggest these data be included and discussed for a more coherent and strong publication (if the manuscript can be reworked for publication based on revision of the above-mentioned flaws).

*RE: It would be more informative to have more data, for example, the forest floor C and N stocks, although it is quite thin in these site that we used here. This is also typical in our area in subtropical area. Regarding the litterfall data, it was measured for other experiment and the data is not complete if we use it for our purpose as they only measured these three forests of 24 year old, instead of all the three age classes. For the fine root data, we recently developed one MS and submitted it to BMC Ecology recently.*

(6) The language of the manuscript needs substantial linguistic checking by a native English speaker. The wording is not correct in many places and several sentences are hard to understand.

*RE: We checked the linguistic errors and revised our manuscript intensively. I hope this version is clear and acceptable.*

[revised manuscript text omitted]

带格式式表格

**Table 2** The effects of plantation stand, stand age and soil depth on soil OC concentration, N concentration, OC and N stock. Values shown are ANOVA F values and P values. Bold font indicates significant differences at $p<0.05$.

| Factor | OC concentration | | N concentration | | OC stock | | N stock | |
|---|---|---|---|---|---|---|---|---|
| | F value | P | F value | P | F value | P | F value | P |
| stand | 24.30 | **<0.0001** | 11.89 | **<0.0001** | 36.73 | **<0.0001** | 15.17 | **<0.0001** |
| age | 0.00 | 0.9942 | 60.53 | **<0.0001** | 35.99 | **<0.0001** | 0.54 | 0.5814 |
| depth | 699.57 | **<0.0001** | 205.33 | **<0.0001** | 489.87 | **<0.0001** | 65.62 | **<0.0001** |
| stand×age | 2.07 | 0.1283 | 10.50 | **<0.0001** | 2.87 | **0.0236** | 3.92 | **0.0042** |
| stand×depth | 8.61 | **0.0002** | 2.39 | 0.0937 | 5.53 | **0.0003** | 1.15 | 0.3354 |
| age×depth | 2.00 | 0.1588 | 0.05 | 0.8163 | 16.79 | **<0.0001** | 4.18 | **0.0027** |
| stand×age×depth | 0.15 | 0.8579 | 0.73 | 0.4834 | 1.03 | 0.4142 | 0.67 | 0.7217 |

**Figure Captions**

**Fig. 1** Soil organic carbon (OC) concentration and nitrogen (N) concentration in pure *Pinus*, *Cinnamomum* and mixed *Pinus-Cinnamomum* stands in 0-10 cm, 10-20 cm and 20-30 cm soil depth at the age of 10, 24 and 45 years. Error bars indicate standard errors. Different letters indicate significant differences among different stands within the same soil profile and age stages ($p<0.05$).

**Fig. 2** Total soil organic carbon (OC) and nitrogen(N) stock in pure *Pinus*, *Cinnamomum* and mixed *Pinus-Cinnamomum* stands in 0-30 cm soil depth at the age of 10, 24 and 45 years. Error bars indicate standard errors. Different letters indicate significant differences among different stands within the same ages ($p<0.05$).

**Fig. 3** Soil organic carbon (OC) and nitrogen(N) stock in pure *Pinus*, *Cinnamomum* and mixed *Pinus-Cinnamomum* stands in 0-10 cm, 10-20 cm and 20-30 cm soil depth at the age of 10, 24 and 45 years. Error bars indicate standard errors. Different small letters indicate significant differences among different stands within the same soil profile and age stages ($p<0.05$). Different capital letters indicate significant differences among different soil layers within the same stand at given stand age ($p<0.05$).

**Fig. 4** Investigation of additive or non-additive interactions for soil OC stock (a) and N stock (b) in *Pinus- Cinnamomum* mixed stands in 0-10 cm, 10-20 cm and 20-30 cm soil depth at the age of 10, 24 and 45 years. Observed values were compared to expected values calculated as the average value in monocultures of *Pinus* and *Cinnamomum*. Error bars represent 95% CI, and mixtures for which the CIs do not cross y = 0 are considered to be significantly non-additive. Different letters indicate significant differences among different stand ages within the same soil depth profile.

Fig. 1

[Figure]

☐ *Pinus* stands  ■ *Pinus-Cinnamomum* stands  ☐ *Cinnamomum* stands

Fig. 2

[Figure]

Fig. 3

C stock (Mg ha⁻¹)

N stock (Mg ha⁻¹)

[Figure]

□ *Pinus* stands  ■ *Pinus-Cinnamomum* stands  □ *Cinnamomum* stands

Fig. 4

(a)

[Figure]

(b)

---

## Author Response (AR2)

Corresponding author:

Pifeng Lei

Faculty of Life Science & Technology

Central South University of Forestry & Technology

Address: No. 498 Southern Shaoshan Road

     410004, Changsha, Hunan, China

Tel.:   +86-731-85623458

Fax:   +86-731-85623483

Email: pifeng.lei@outlook.com

Dear Prof. Fontaine,

Thank you very much for your comments. According to your and the reviewer's comment, we added one table of fine root biomass data as supplementary material. Here we selected to show the data of fine root biomass in table rather than graph to avoid the repetition of same results in the same format in two MS, as we developed graph for that in another MS which has been submitted to BMC Ecology recently. Of course it is a brilliant idea to add these data as supplementary material, as we have concerns to report one result twice before. Beside, we also made some minor changes in the MS. We hope this revised version is acceptable for publication and please let us know if there is something further needs to be done. Thank you very much for your patience.

With best regards,

Pifeng Lei

On behalf of co-authors

**Responses to Reviewer**

I think that the authors make a good job in order to taking into account the majority of the comments by the 2 reviewers.

In my point of view the paper is acceptable for publication. But I think that the information about roots will be improve the story of this experimentation. I know that these data are the main core of your next paper (see your comments "Regarding the fine root data, we have developed a manuscript for that and submitted to BMC Ecology recently. See P8, L1-3."), but for me the roots biomass data merit one graph or table in supp mat.

**RE:** *Thank very much for your positive feedback. It is a very good suggestion to show the fine root biomass in supplementary material. Accordingly, we developed one table of fine root biomass data in supplement, rather than graph in order to avoid the simple repetition, as it was shown in graph for that in another MS which has been submitted to BMC Ecology recently.*

[revised manuscript text omitted]

Bar chart figure. Top panel: C Stock (Mg ha$^{-1}$) on y-axis (0 to 60) for 10 years, 24 years, and 45 years. Three bars each for Pinus stands, Pinus-Cinnamomum stands, and Cinnamomum stands with significance letters. Bottom panel: N Stock (Mg ha$^{-1}$) on y-axis (0 to 6) for same age groups.

■ *Pinus* stands  ■ *Pinus-Cinnamomum* stands  ☐ *Cinnamomum* stands

Fig. 3

[Figure]

C stock (Mg ha⁻¹)

N stock (Mg ha⁻¹)

10 years

24 years

45 years

☐ *Pinus* stands   ■ *Pinus-Cinnamomum* stands   ☐ *Cinnamomum* stands

Fig. 4

[Figure]